# Deciphering the mechanism of the Ni-photocatalyzed C–O cross-coupling reaction using a tridentate pyridinophane ligand

Hanah Na [1] & Liviu M. Mirica [1✉]

Photoredox nickel catalysis has emerged as a powerful strategy for cross-coupling reactions. Although the involvement of paramagnetic Ni(I)/Ni(III) species as active intermediates in the catalytic cycle has been proposed, a thorough spectroscopic investigation of these species is lacking. Herein, we report the tridentate pyridinophane ligands $^{R}$N3 that allow for detailed mechanistic studies of the photocatalytic C–O coupling reaction. The derived ($^{R}$N3)Ni complexes are active catalysts under mild conditions and without an additional photocatalyst. We also provide direct evidence for the key steps involving paramagnetic Ni species in the proposed catalytic cycle: the oxidative addition of an aryl halide to a Ni(I) species, the ligand exchange/transmetalation at a Ni(III) center, and the C–O reductive elimination from a Ni(III) species. Overall, the present work suggests the $^{R}$N3 ligands are a practical platform for mechanistic studies of Ni-catalyzed reactions and for the development of new catalytic applications.

[1] Department of Chemistry, University of Illinois at Urbana-Champaign, 600 S. Mathews Avenue, Urbana, IL 61801, USA. ✉email: mirica@illinois.edu

The carbon–oxygen (C–O) bond-forming reactions are of particular interest in the field of organic synthesis, as C–O bonds are prevalent in many natural products, pharmaceuticals, and agrochemicals[1–3]. Although Pd-based C–O cross-coupling catalysts are well established[4–7], there has been a great interest in developing sustainable catalytic systems based on earth-abundant nickel. In this regard, photoredox/nickel dual catalysis and light-promoted Ni catalysis have emerged as powerful strategies for challenging C–O bond formations[8–11]. Accordingly, an in-depth mechanistic understanding of Ni-mediated photocatalysis is essential for rational reaction design and optimization. From a mechanistic viewpoint, while the photocatalytic cycle is well-understood, the Ni-mediated redox cycle remains elusive as intermediates in various oxidation states (from $Ni^0$ to $Ni^{IV}$) have been proposed[12]. For example, the involvement of paramagnetic $Ni^I$ and $Ni^{III}$ species has been commonly implicated, yet such intermediates have not been thoroughly investigated, and the key catalytic steps of oxidative addition, transmetalation, and reductive elimination have rarely been observed directly[10,13,14].

Our group has employed tetradentate azamacrocycle $N,N'$-dialkyl-2,11-diaza[3.3](2,6)pyridinophanes ($^RN4$) ligands to isolate and characterize $Ni^{III}$ species capable of C–C[15–18] and C–heteroatom[15,19,20] bond formation reactions, and to probe the involvement of high-valent Ni species in these reactions. Since the most ubiquitous ligands in Ni cross-coupling and photocatalytic reactions are the bidentate bipyridine ligands[8–11,21], the $^RN4$ ligands were thought to be less effective ligands for catalytic applications due to the crowded environment around the Ni center, which is also expected to disfavor the formation of $Ni^I$ species. By contrast, a bidentate ligand structure is not suitable for stabilizing high-valent Ni species, thus hampering the investigation of such Ni intermediates[22]. Considering these aspects, we sought to identify an optimal ligand framework positioned in between bipyridine and $^RN4$ ligands in terms of denticity, molecular structure, and functionality, and thus the pyridinophane tridentate $^RN3$ ligands were targeted (Fig. 1a). These $^RN3$ ligands are structurally analogous to the $^RN4$ ligands, possessing a rigid aromatic pyridinophane framework and containing only one flexible chelating arm that allows for either a $\kappa^2$ or $\kappa^3$ coordination. Surprisingly, such $^RN3$ ligands have never been synthesized to date, likely due to the lack of an efficient synthetic route. Several other well-known *fac*-capping tridentate N-donor ligands such as tris(pyrazoyl)borate ($Tp^-$)[22], tris(pyrazolyl) methane (Tpm)[23], or 1,4,7-trimethyl-1,4,7-triazacyclononane (Me₃TACN, Fig. 1b)[24] have been reported to stabilize high-valent organometallic Ni complexes, or even an organometallic $Ni^I$ complex[25], yet these ligands do not seem to stabilize mononuclear $LNiX_2$ (X = halide) complexes, dinuclear halide-bridged or homoleptic 2:1 L:Ni complexes being generated instead[26–28].

In this work, we report the development of $N$-alkyl-2-aza[3.2] (2,6)pyridinophanes ($^RN3$, R = Me or $^iPr$) ligands, which embody our design criteria for photocatalytic applications and mechanistic studies. The corresponding ($^RN3$)Ni complexes are active C–O cross-coupling catalysts upon light excitation and without an added precious-metal photocatalyst. Importantly, we show that the utilization of the tridentate $^RN3$ ligands allows for a comprehensive examination of the Ni intermediates proposed in the individual steps of the cross-coupling catalytic cycle: oxidative addition, transmetalation/ligand exchange, and reductive elimination, all these steps involving paramagnetic $Ni^I$ or $Ni^{III}$ species.

## Results and discussion

### Ligand development
In 1958, Overberger et al. reported the reduction of N-nitrosodibenzylamines by $Na_2S_2O_4$ to generate the hydrocarbon product with evolution of $N_2$[29]. However, this reaction has received little attention since then, and only one report by Takemura et al. in 1988 exploited the $N$-nitroso reduction reaction to obtain [2.2]cyclophane derivatives[30]. We speculated that this N-extrusion reaction would provide an efficient synthetic pathway for tridentate pyridinophane ligands. Our synthesis starts with the unsymmetrical pyridinophane precursor $N$-tosyl-2,11-diaza[3.3](2,6)pyridinophane ($^{TsH}N4$, Fig. 2a)[31]. For the nitrosylation step, $^{TsH}N4$ was treated with $HNO_2$ to generate the $N$-nitroso intermediate $^{TsNO}N4$ in 85% yield. Subsequent reductive elimination of nitrogen from the N-nitrosoamine was performed with $Na_2S_2O_4$ under basic conditions, converting $^{TsNO}N4$ to $^{Ts}N3$ in 75% yield. Additional steps of detosylation and methylation or isopropylation afforded the alkylated ligand variants $^{Me}N3$ or $^{iPr}N3$ in ~45% overall yields, confirming that further functionalization can be easily achieved. The $N$-nitroso reduction conditions are mild, suggesting a broad functional group tolerance. Since all reaction steps do not require extensive purification and their yields are high, the development of this synthetic route provides an opportunity to access a variety of ligand structures by further functionalization or starting from different secondary amine precursors. Notably, Levin et al. have recently reported the development of an anomeric amide reagent for the oxidative extrusion of nitrogen from dibenzylamines via a isodiazene intermediate to yield C–C coupled products[32]. By comparison, our approach employs a reduction of the $N$-nitrosoamine intermediate and only uses simple and commonly available reagents, and thus allowing for bis-2-picolylamines to be used as substrates.

### Synthesis and characterization of ($^RN3$)NiCl₂ complexes
The reaction of $^{Me}N3$ or $^{iPr}N3$ with Ni(DME)Cl₂ in CH₂Cl₂ generated the green $Ni^{II}$-dichloride complexes ($^{Me}N3$)NiCl₂ (**1a**) and ($^{iPr}N3$)NiCl₂ (**1b**), respectively (Fig. 2b). This is in contrast to other tridentate N-donor ligands such as $Tp^-$, Tpm, or Me₃TACN, which tend to form dinuclear halide-bridged or homoleptic 2:1 L:Ni complexes instead of $LNiCl_2$ complexes[26–28]. Both complexes **1a** and **1b** are paramagnetic, as shown by the paramagnetic NMR spectrum of **1b** (Supplementary Fig. 11), and the effective magnetic moment $\mu_{eff}$ of 3.03 $\mu_b$ determined using the Evans method is consistent with a triplet ground state ($S = 1$), as expected for a high-spin $Ni^{II}$ center[33,34]. The obtained magnetic moment value is similar to those of triplet ground state Ni(bpy)X₂ (X = halide) complexes[35], although these complexes adopt a tetrahedral geometry[36]. The molecular structures of **1a** and **1b** were determined by single crystal X-ray diffraction (Fig. 2c), to reveal similar coordination environments around the Ni center, except that the structure of **1b** exhibits slight distortions due to the bulkier $^iPr$ substituent (Supplementary Fig. 55). The structure of **1a** reveals a five-coordinate $Ni^{II}$ center in a square pyramidal geometry with a calculated trigonality index parameter $\tau$ of 0[37], with an average Ni–$N_{py}$ bond length of 2.058 Å and an axial Ni–$N_{amine}$ bond length of 2.056 Å, which are similar to those of the related complex ($^{tBu}N4$)NiCl₂[38]. A closer look at the space-filling representation of **1a**, as shown from the bottom view and along the Ni–$N_{amine}$ axial axis (Fig. 2d), reveals that the Ni center is sterically protected by the H atoms of the ethylene bridge and thus could lead to an increased stability of the high- or low-valent Ni intermediates (see below).

The electrochemical properties of ($^RN3$)NiCl₂ complexes were investigated by cyclic voltammetry (CV, Fig. 3c, Supplementary Figs. 13 and 14). The CVs of **1a** and **1b** in MeCN reveal a pseudoreversible oxidation wave at 0.71 V vs $Fc^{+/0}$, which is assigned to the $Ni^{III/II}$ couple, as well as an irreversible reduction wave at −1.20 V vs $Fc^{+/0}$ for **1a** and −1.22 V for **1b**, followed by

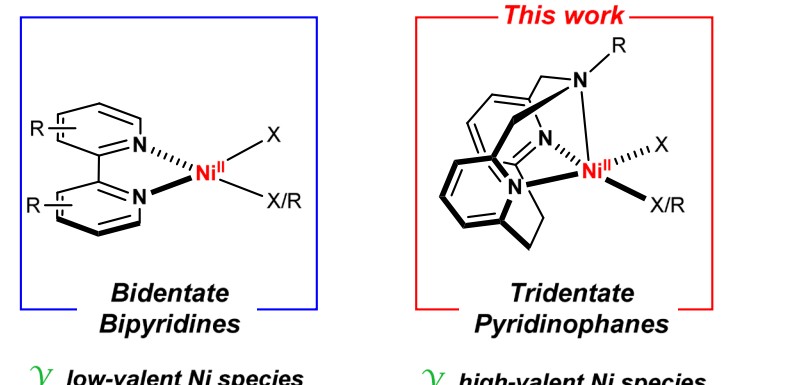

*a) Tridentate ligand framework design*

**Bidentate Bipyridines**

✓ low-valent Ni species
✗ high-valent Ni species

**Tridentate Pyridinophanes**

✓ high-valent Ni species
✓ low-valent Ni species

**Tetradentate Pyridinophanes**

✓ high-valent Ni species
✗ low-valent Ni species
✗ structurally crowded

*b) Examples of tridentate ligands*

**Commonly used tridentate ligands**

Tpm        Tp⁻        R₃TACN

**This work — $^{R}$N3 ligands**

$^{R}$N3

**Fig. 1 Comparison of different N-donor ligands. a** Tridentate ligand framework design. **b** Representative tridentate N-donor ligands commonly used for stabilizing uncommon Ni oxidation states and the $^{R}$N3 tridentate ligands developed in this work.

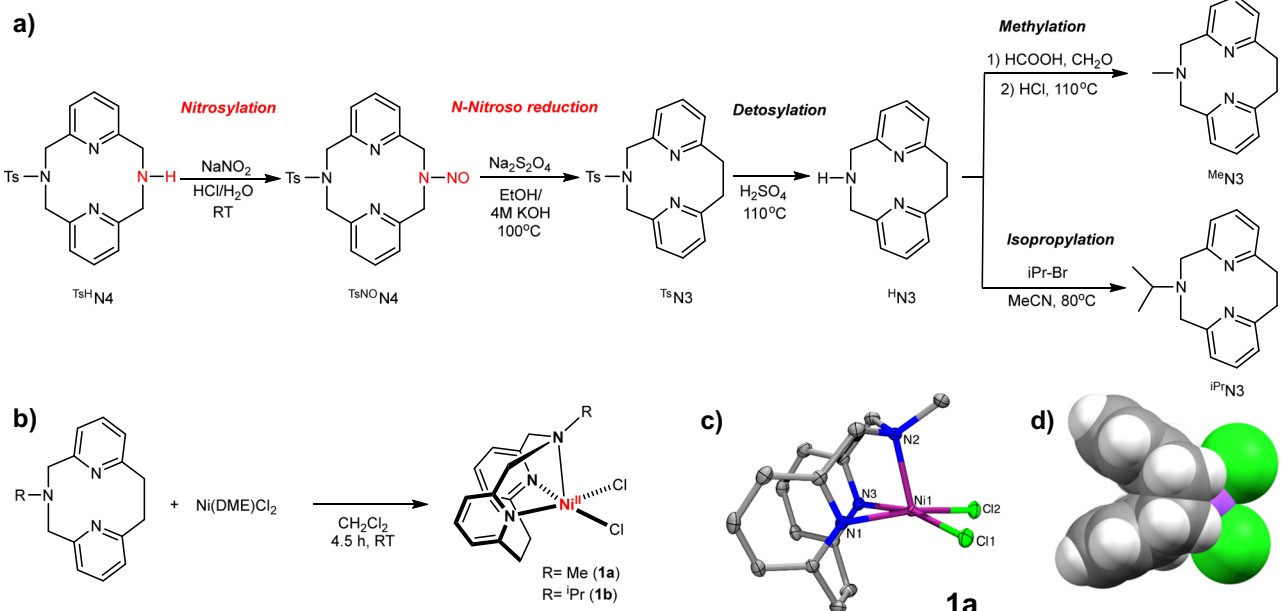

**Fig. 2 Synthesis and structural characterization. a** Synthesis of $^{R}$N3 pyridinophane ligands. **b** Synthesis of ($^{R}$N3)NiCl₂ complexes (1a and 1b) and **c** ORTEP representation of the 1a obtained by X-ray diffraction (ellipsoids are shown at 50% probability and the hydrogen atoms omitted for clarity). Selected bond lengths (Å): Ni–N$_{py}$ 2.058, Ni–N$_{amine}$ 2.056, Ni–Cl 2.322. **d** Space-filling model of 1a highlighting the steric protection of the Ni center by the H atoms of the ethylene bridge.

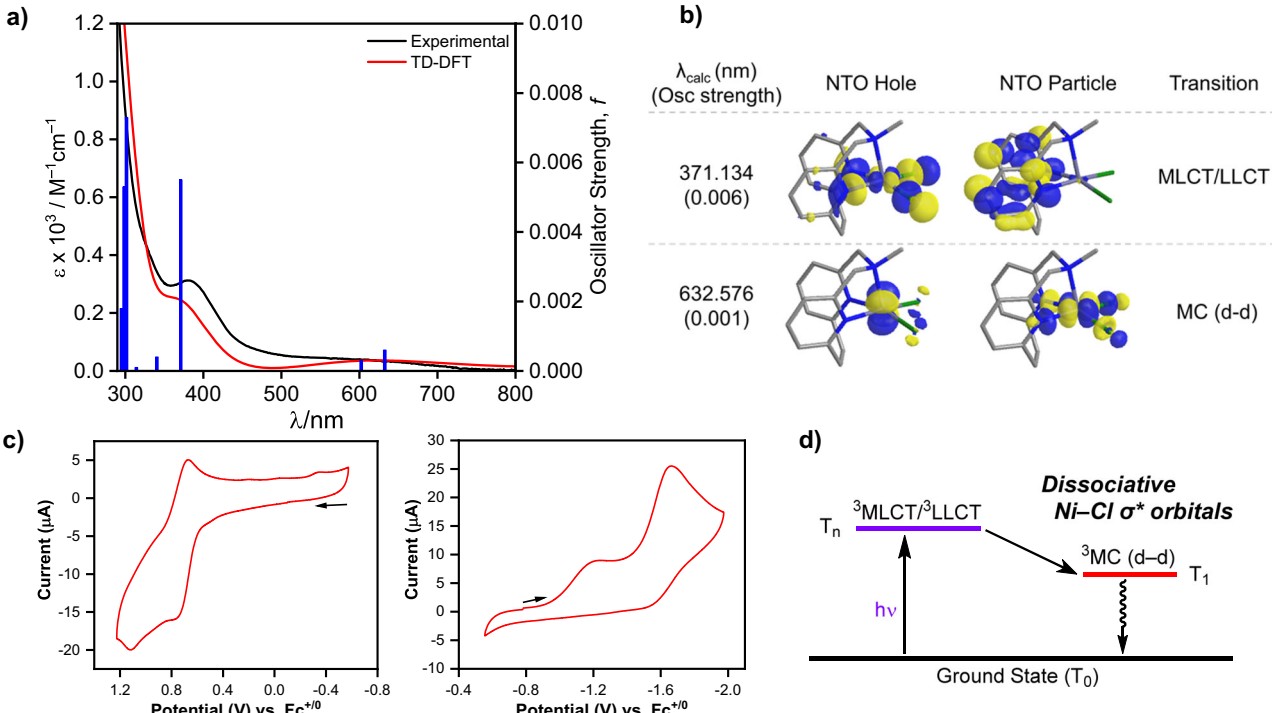

**Fig. 3 Characterization of 1a. a** Experimental (black line) and simulated (red line, using Gaussian peak shapes with a full width at half maximum FWHM of 0.33 eV) UV–vis spectrum of 1a with overlaid relevant oscillator strengths (solid blue bars) from TD–DFT calculations. **b** Natural transition orbitals (NTOs, 0.05 isocontour value) associated with the visible absorption bands of 1a. **c** Cyclic voltammograms (CVs) of 1a in 0.1 M TBAPF$_6$/MeCN (scan rate 0.1 V/s). **d** A simplified excited-state diagram for 1a, showing the population of the lower energy excited state, a $^3$MC d-d state.

a pseudoreversible reduction wave at $-1.60$ V vs Fc$^{+/0}$ for **1a** and $-1.72$ V for **1b**, respectively. The first reduction wave is proposed to correspond to the conformation of the Ni complex in which the $^R$N3 ligand adopts a $\kappa^2$ binding mode, in line with the flexibility of the pyridinophane ligands observed previously[24,39]. The electrochemical irreversibility of the second reduction event is likely due to the loss of a Cl$^-$ ligand and the change in coordination geometry from square pyramidal to tetrahedral upon formation of the Ni$^I$ species (i.e., an EC mechanism: an electron transfer followed by a chemical reaction)[40]. It is important to note that the oxidation and reduction potentials for **1a** and **1b** are similar to those of (dtbbpy)NiCl$_2$, which exhibits an oxidation wave at 0.56 V vs Fc$^{+/0}$ and an irreversible reduction wave at $-1.37$ V vs Fc$^{+/0}$, followed by a pseudor-eversible reduction wave at $-1.74$ V vs Fc$^{+/0}$ (Supplementary Fig. 16). By comparison, while for ($^{Me}$N4)NiCl$_2$ the Ni$^{III/II}$ oxidation event was observed at 0.52 V vs Fc$^{+/0}$, no reduction event was observed up to $-2.00$ V vs Fc$^{+/0}$[41], supporting our hypothesis that tetradantate ligands cannot stabilize Ni$^I$ species to a large extent. The accessible reduction events observed for ($^R$N3)NiCl$_2$ suggest that for these complexes the Ni$^I$ oxidation state is accessible via one-electron reduction with a chemical reductant and thus it can be exploited in reactivity and mechanistic studies. When compared to (dtbbpy)NiCl$_2$, the ($^R$N3)NiCl$_2$ complexes exhibit similar oxidation and reduction potentials, which is essential for photocatalysis, yet they possess a structural advantage by containing the flexible axial amine arm that is beneficial for characterization of reaction intermediates (see below).

The UV-visible absorption spectra of **1a** and **1b** were then obtained and also analyzed computationally. The UV-vis absorption spectrum of **1a** shows an intense band around 380 nm and a less intense band around 640 nm (Fig. 3a). To probe the nature of the electronic transitions, time-dependent

density functional theory (TD-DFT) calculations were performed. As the ground state of **1a** has triplet spin multiplicity, the higher energy absorption feature (380 nm) is attributed to a combination of a spin-allowed Cl → $^R$N3 ligand-to-ligand charge transfer ($^3$LLCT) and a Ni → $^R$N3 metal-to-ligand charge transfer ($^3$MLCT), while the lower energy absorption band (640 nm) is attributed to a metal-centered ($^3$MC) d-d electronic transition (Supplementary Figs. 63–66). The energy of the $^3$MC d-d transition (640 nm ≈ 1.93 eV) is consistent with the electro-chemical gap observed by CV (0.71 V $-$ ($-1.20$ V) = 1.91 eV), further supporting that the lowest excited state (T$_1$) has primarily MC d–d character. For a qualitative representation of the specified transitions between the ground and excited states, we performed a natural transition orbital (NTO) analysis that provides a localized picture of the transition density matrix (Fig. 3b)[42]. The NTO analysis supports the assignment of the 380 nm transition to a T$_n$ excited state with a mixed LLCT/MLCT character, while the 633 nm transition involves an electron excitation between predominantly metal-based orbitals, thus indicating a MC (d–d) transition. For 3d transition metal complexes the MC (d–d) state usually lies lower in energy than the CT state(s), due to the intrinsically low ligand field strength, and thus resulting in the relaxation of the CT state to the lower-lying MC (d–d) state[43,44]. Therefore, we propose that on initial population of the MLCT/LLCT state is followed by fast relaxation to the lower MC (d–d) state with an appreciable Ni–Cl σ* orbital character (Fig. 3d), which then promotes a metal–ligand homolytic bond cleavage[45,46], as observed recently by Doyle et al. for a similar Ni system[47]. Since **1a** shows a prominent absorption band at 380 nm, a purple LED lamp ($\lambda_{max} = 390$ nm) was chosen as a light source for the direct light-promoted C–O coupling reaction (see below). Finally, the UV–vis absorption spectrum and TD-DFT calculation results for **1b** are similar to

**Table 1 Light-promoted Ni catalytic reaction development and optimization[a].**

| Entry | Variation from the standard conditions | Yield (%) |
|---|---|---|
| 1 | None | 95 |
| 2 | No light (dark) | 0 |
| 3 | No ($^{Me}$N3)NiCl$_2$ | 0 |
| 4 | No base | 0 |
| 5 | Blue LED instead of purple LED | 0 |
| 6 | DABCO instead of quinuclidine, 24 h | 77 |
| 7 | DABCO instead of quinuclidine, 36 h | 96 |
| 8 | K$_2$CO$_3$ instead of quinuclidine | 0 |
| 9 | ($^{Me}$N3)NiCl$_2$ 0.2 mol% instead of 2 mol% | 90 |
| 10 | ($^{Me}$N3)NiBr$_2$ instead of ($^{Me}$N3)NiCl$_2$ | 90 |
| 11 | ($^{iPr}$N3)NiCl$_2$ instead of ($^{Me}$N3)NiCl$_2$ | 82 |
| 12 | $^{Me}$N3 (2 mol%) + Ni(DME)Cl$_2$ (2 mol%) instead of ($^{Me}$N3)NiCl$_2$ | 77 |
| 13 | ($^{Me}$N4)NiCl$_2$ instead of ($^{Me}$N3)NiCl$_2$ | 2 |

[a]Reaction conditions: In a N$_2$-filled glovebox, 4-bromoacetophenone (0.4 mmol, 1.0 equiv), quinuclidine (0.44 mmol, 1.1 equiv), MeOH (1.6 mmol, 4 equiv), ($^{Me}$N3Ni)Cl$_2$ (2 mol%) and a magnetic stir bar were added into 0.4 mL THF in a vial. The vial was irradiated with one purple LED lamp (52 W, 390 nm) under fan cooling. After 24 h, 1,3-benzodioxole (0.4 mmol) was added to the reaction mixture as a standard and the residue was analyzed by $^1$H NMR and GC-FID to determine the yield of C–O coupled product.

those for **1a**, indicating a minimal effect of the N-substituent on the electronic structure (Supplementary Figs. 67–72).

**Photocatalytic C–O coupling reactions mediated by ($^R$N3) NiCl$_2$ complexes.** To probe the relevance of the ($^R$N3)NiCl$_2$ complexes in photocatalysis, we evaluated their light-promoted catalytic activity in C–O cross-coupling without any additional photocatalyst. Our initial efforts focused on optimizing the reaction conditions using methanol and 4-bromoacetophenone (Table 1 and Supplementary Tables 2–8). In the presence of 2 mol% ($^{Me}$N3)NiCl$_2$ (**1a**) and quinuclidine in THF, the desired C–O coupled product 4-methoxyacetophenone was obtained in 95% yield under purple LED ($\lambda_{max}$ = 390 nm) irradiation (Table 1, entry 1). Control experiments revealed that the reaction did not proceed without Ni catalyst, light, or base (Table 1, entries 2–4). The desired product was not observed when the reaction was irradiated with blue LED, suggesting the crucial role of the excitation energy (Table 1, entry 5). While inorganic bases such as K$_2$CO$_3$ were not effective, possibly due to the precipitation of halides required to stabilize the Ni$^I$ intermediate[14] or the unproductive decomposition of the Ni$^{III}$ intermediates (Supplementary Figs. 46–47), DABCO was effective but proceeded at a slightly slower rate and thus requiring a longer reaction times (Table 1, entries 6–8). Switching from **1a** to **1b** or ($^{Me}$N3)NiBr$_2$, lowering of catalyst loading to 0.2 mol%, or the addition of $^{Me}$N3 and Ni(DME)Cl$_2$ separately instead of **1a** gave comparable yields (Table 1, entries 9–12). Notably, a 0.2 mol% catalyst loading is significantly less (~25–50 fold decrease) than the loadings used in other Ni-photocatalyzed C–O cross-couplings[10,11]. Moreover, other alcohols such as 1-hexanol and benzyl alcohol could also be employed (Supplementary Table 7, 86–90% product yield). The combination of **1a** and heterogenous Zn$^0$ or Mn$^0$ without light was ineffective in our conditions, likely due to the slow electron transfer kinetics or insufficient generation of the active species during catalysis[48,49]. Interestingly, the use of a Ni complex supported by another tridentate N-donor ligand 1,4,7-triosopropyl-1,4,7-triazacyclononane, (iPr$_3$TACN)Ni$^{II}$Cl$_2$, generated the desired product only in a moderate yield (51%, Supplementary Table 8),

while the ($^R$N4)Ni$^{II}$Cl$_2$ complex supported by the tetradentate ligand $^{Me}$N4 yielded a negligible amount of product, validating our ligand design principle (Table 1, entry 13).

The same C–O coupling reaction was also probed under common dual Ni/photoredox catalytic conditions using various Ir or Ru photocatalysts and blue LED light ($\lambda_{max}$ = 456 nm) (Table 2). The photocatalyst screening experiments revealed that the Ir photocatalysts tested here gave the desired C–O coupled product in 80–98% yield, while the Ru catalysts showed no reactivity.

**EPR and reactivity studies.** Several reports have proposed the oxidative addition of an organic halide to Ni$^I$ to generate a Ni$^{III}$ species is a key step in the C–O cross-coupling catalytic cycle[13,14,47], yet minimal evidence was provided and such a step has been underexplored experimentally[50,51]. Our Ni catalysts have accessible Ni$^{II/I}$ redox potentials, thus we aimed to probe the reduction of Ni$^{II}$ to Ni$^I$ and the subsequent oxidative addition of an aryl halide to generate a Ni$^{III}$ intermediate by using X-band electron paramagnetic resonance (EPR) spectroscopy. For the EPR studies, **1b** was used due to its better solubility vs **1a**. Upon the reduction of **1b** by 1 equiv decamethylcobaltocene (CoCp*$_2$) at −95 °C (Fig. 4a), a Ni$^I$ species **2** was formed that exhibits a rhombic EPR spectrum with $g_x$ = 2.241, $g_y$ = 2.113, and $g_z$ = 2.049 (Fig. 4d), and suggesting the presence of a metal-based radical. The detection of an EPR-active mononuclear Ni$^I$ species is important since such an intermediate is essential for an efficient oxidative addition step. By comparison, in the catalytic systems involving bidentate bipyridine ligands relatively stable dinuclear Ni$^I$ species are formed as side products that hinder catalysis, especially at higher substrate concentrations and catalyst loadings[13,47]. In our case, the Ni$^I$ species **2** decays rapidly to EPR inactive species in the absence of substrate, yet no dinuclear Ni$^I$ species are formed, suggesting that the tridentate $^R$N3 ligand might prevent the formation of such catalytically inactive species due to the steric hindrance provided by the axial amine arm of the $^R$N3 ligand (see Supplementary Figs. 58 and 59 for the space filling

**Table 2 Dual Ni/photoredox catalytic reaction optimization[a].**

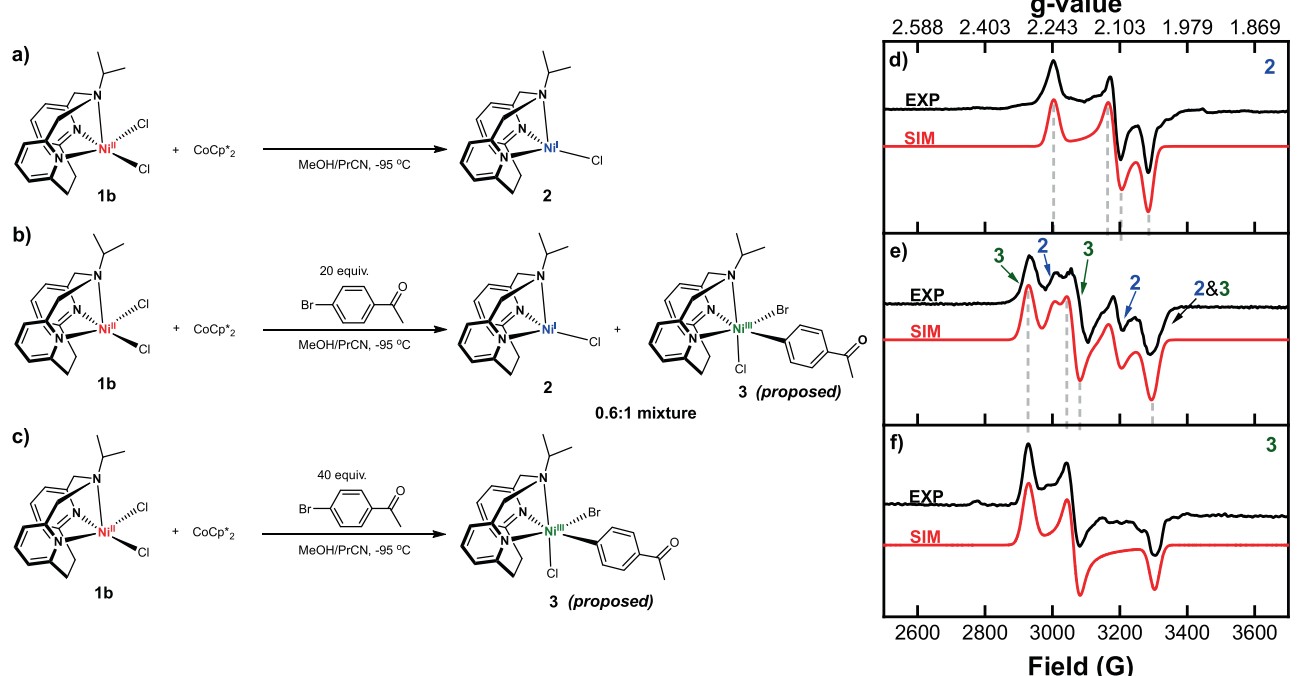

| Entry | Photocatalyst | Yield (%) |
|---|---|---|
| 1 | No photocatalyst | 0 |
| 2 | [Ir(dF(CF$_3$)$_2$ppy)(dtbbpy)]PF$_6$ | 86 |
| 3 | Ir(ppy)$_3$ | 90 |
| 4 | [Ir(ppy)$_2$(dtbbpy)]PF$_6$ | 98 |
| 5 | [Ru(phen)$_3$]Cl$_2$ | ~1% |
| 6 | [Ru(bpz)$_3$](PF$_6$)$_2$ | 0 |

[a]Reaction conditions: In a N$_2$-filled glovebox, 4-bromoacetophenone (0.4 mmol, 1.0 equiv), quinuclidine (0.44 mmol, 1.1 equiv), MeOH (1.6 mmol, 4 equiv), ($^{Me}$N3)NiCl$_2$ (2 mol%), photocatalyst (0.5 mol%) and a magnetic stir bar were added into 0.4 mL THF in a vial. The vial was irradiated with one blue LED lamp (50 W, 456 nm) under fan cooling. After 24 h, 1,3-benzodioxole (0.4 mmol) was added to the reaction mixture as a standard and the residue was analyzed by $^1$H NMR and GC-FID to determine the yield of C–O coupled product.

**Fig. 4 EPR studies of the Ni$^I$ species. a–c** (left) Experimental setup for EPR detection of proposed Ni$^I$ and Ni$^{III}$ species upon reduction of 1b and subsequent oxidative addition of aryl bromide. (right) Experimental (black) and simulated (red) EPR spectra of the reaction mixtures recorded at 77 K in 1:5 MeOH:PrCN frozen glass: **d** 1b + CoCp*$_2$, **e** 1b + CoCp*$_2$ + 20 equiv 4-bromoacetophenone, **f** 1b + CoCp*$_2$ + 40 equiv 4-bromoacetophenone. The following parameters were used for simulation of **d** $g_x = 2.241$, $g_y = 2.113$, $g_z = 2.049$; **f** $g_x = 2.298$, $g_y = 2.197$, $g_z = 2.037$; **e** the simulations in **d** and **f** were added in a 0.6:1 ratio. The structure proposed for species 3 is tentative.

models of **1a** and **1b**). This behavior might explain why higher concentrations of substrate and catalyst can be employed herein, which is in contrast to the conditions utilized in other reported C–O cross-coupling reactions[10,11,14]. When 20 equiv 4-bromoacetophenone were added to a thawing solution of **2** (Fig. 4b), the resulting EPR spectrum could be simulated with two sets of parameters corresponding to two different species in a 0.6:1 ratio and that were tentatively assigned to the initial Ni$^I$ species and a new Ni$^{III}$ species (Fig. 4e and Supplementary Fig. 32). We tentatively propose that the Ni$^{III}$ species is a transient 6-coordinate ($^{iPr}$N3)Ni$^{III}$(PhAc)BrCl (**3**) complex generated through the oxidative addition of 4-bromoacetophenone to **2**. Moreover, when 40 equiv 4-bromoacetophenone were added to **2** (Fig. 4c), only a rhombic EPR signal with $g_x = 2.298$, $g_y = 2.197$,

and $g_z = 2.037$ formed (Fig. 4f), which was assigned to the proposed structure **3** and thus strongly supporting a direct conversion from Ni$^I$ (**2**) to Ni$^{III}$ (**3**) that is promoted by the excess aryl halide. The observed EPR spectrum for **3** decays quickly upon warming up the solution for seconds at −80 °C (Supplementary Fig. 33), supporting the increased reactivity of **3** and its proposed involvement in catalysis. Finally, while the exact structure of this species **3** is not unambiguously known, it is important to note that to the best of our knowledge the direction observation of the oxidation addition of an aryl halide to a Ni$^I$ species to generate a Ni$^{III}$ species has not been reported to date.

To further probe the role of various Ni$^{III}$ species in the C–O coupling reaction, we independently synthesized the ($^{iPr}$N3)Ni$^{II}$(PhAc)Br complex **4** via oxidative addition of 4-bromoacetophenone

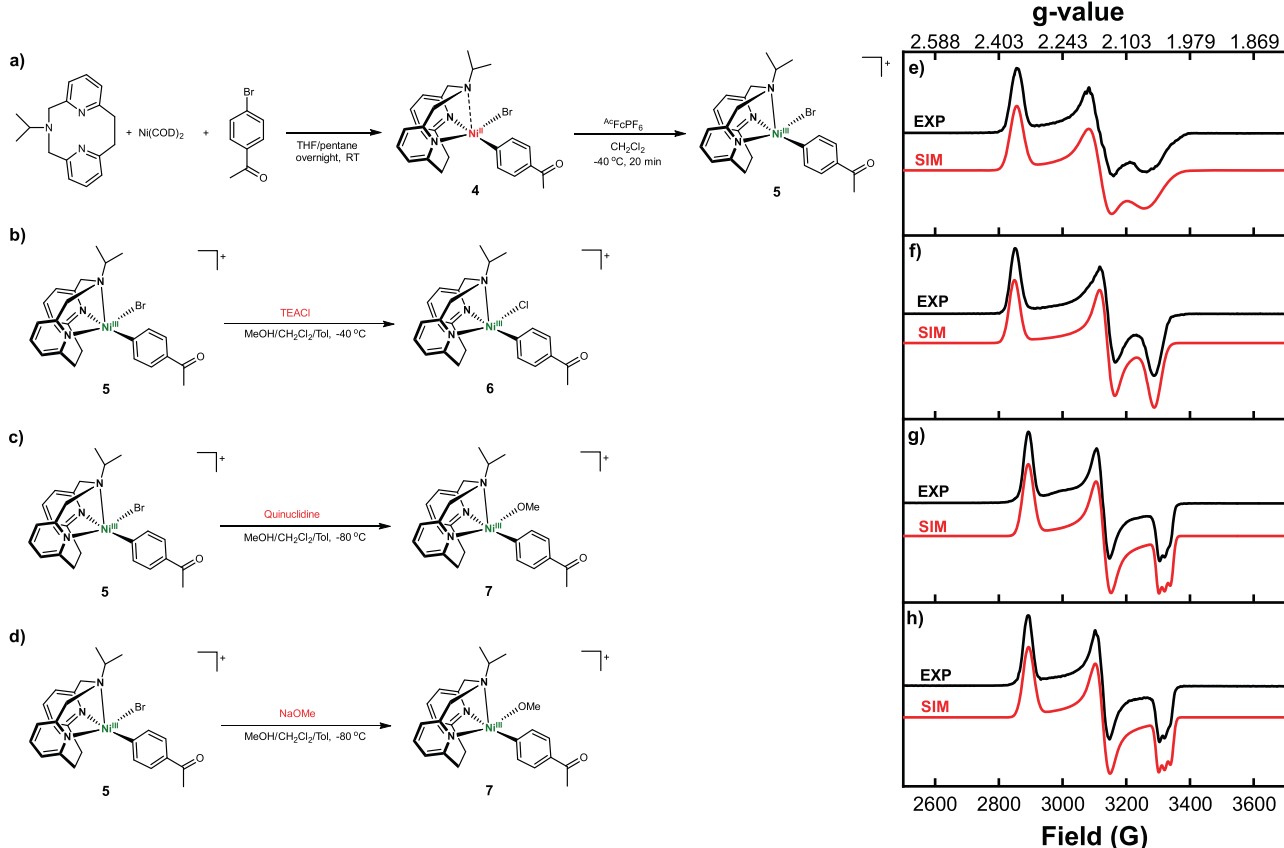

**Fig. 5 EPR studies of Ni$^{III}$ species.** (left) Experimental setup for the synthesis of ($^{iPr}$N3)Ni$^{II}$(PhAc)Br (4) and EPR detection of **a** [($^{iPr}$N3)Ni$^{III}$(PhAc)(Br)]$^{+}$ (5), **b** [($^{iPr}$N3)Ni$^{III}$(PhAc)(Cl)]$^{+}$ (6), and **c** and **d** [($^{iPr}$N3)Ni$^{III}$(PhAc)(OMe)]$^{+}$ (7). (right) Experimental (black) and simulated (red) EPR spectra of the reaction mixtures recorded at 77 K in 1:1:3 MeOH:CH$_2$Cl$_2$:Toluene frozen glass: **e** [($^{iPr}$N3)Ni$^{III}$(PhAc)(Br)]$^{+}$ (5), **f** [($^{iPr}$N3)Ni$^{III}$(PhAc)(Cl)]$^{+}$ (6), and **g** and **h** [($^{iPr}$N3)Ni$^{III}$(PhAc)(OMe)]$^{+}$ (7). The following parameters were used for simulation of **e** $g_x = 2.356$, $g_y = 2.158$, $g_z = 2.059$; **f** $g_x = 2.363$, $g_y = 2.144$, $g_z = 2.047$; **g** $g_x = 2.326$, $g_y = 2.151$, $g_z = 2.027$ (A$_z$(N) = 19.4 G); **h** $g_x = 2.326$, $g_y = 2.152$, $g_z = 2.027$ (A$_z$(N) = 19.5 G).

to Ni(COD)$_2$ in the presence of $^{iPr}$N3 (Fig. 5a). While **4** is unstable in solution at room temperature, rapidly decomposing to the green ($^{iPr}$N3)Ni$^{II}$Br$_2$ complex, it is stable at −40 °C for several hours. The decomposition of a related nickel complex, (bpy)Ni$^{II}$PhBr, to (bpy) Ni$^{II}$Br$_2$ has been observed previously and proposed to proceed via a bimolecular pathway[52,53]. The instability of **4** in solution prevented us from obtaining X-ray quality crystals, nonetheless since **4** is diamagnetic its structural evidence was provided by $^1$H NMR obtained at −40 °C, which shows a slight peak broadening due to the flexible amine arm (Supplementary Fig. 2)[24]. Oxidation of **4** with acetylferrocenium hexafluorophosphate ($^{Ac}$FcPF$_6$) at −40 °C generates the Ni$^{III}$ [($^{iPr}$N3)Ni$^{III}$(PhAc)(Br)]$^{+}$ complex **5** (Fig. 5a). The effective magnetic moment $\mu_{eff}$ of 1.98 $\mu_b$, determined using the Evans method, is consistent with a S = 1/2 ground state for **5**, as expected for a Ni$^{III}$ center[33,34]. The rhombic EPR spectrum of **5** with $g_x = 2.356$, $g_y = 2.158$, $g_z = 2.059$ is significantly different than that of **3** (Fig. 5e vs. Fig. 4f), solidifying the proposed 6-coordinate structure for **3** in Fig. 4c. Complex **5** was stable in solution at −35 °C and can be isolated as a red solid, yet it was recalcitrant to form single crystals. Therefore, the presence of a Ni$^{III}$ center was further confirmed by X-ray photoelectron spectroscopy (XPS), which shows an increase of the Ni 2p3/2 and 2p1/2 binding energies of ~1.5 eV for **5** versus **4**, demonstrating the presence of a more oxidized Ni center (Supplementary Fig. 60)[54].

The appreciable stability of **5** allowed us to investigate its reactivity. When **5** was reacted with 1 equiv tetraethylammonium chloride (TEACl, Fig. 5b), a rhombic EPR spectrum with

$g_x = 2.363$, $g_y = 2.144$, $g_z = 2.047$ and sharper signals than the spectrum of **5** was obtained, indicating the likely formation of the [($^{iPr}$N3)Ni$^{III}$(PhAc)(Cl)]$^{+}$ species **6** (Fig. 5f vs. Fig. 5e). A similar EPR signal broadening due to superhyperfine coupling to the Br atom ($I = 3/2$) vs the Cl atom ($I = 3/2$) was observed in a previous study by our group, where the structural analysis of [($^{tBu}$N4)Ni$^{III}$Ar(X)]$^{+}$ complexes (X = Br or Cl) confirmed the effect of the halide ligand onto the corresponding EPR spectra[15]. Importantly, an EPR spectrum resembling the proposed 6-coordinate species ($^{iPr}$N3)Ni$^{III}$(PhAc)BrCl (**3**) was not observed, supporting the instability of this sterically hindered species. GC-MS analysis of the solution of **6** after warming up to room temperature in the absence of MeOH reveals Ar–Cl product formation (Supplementary Fig. 37), while the reaction between ($^{iPr}$N3)Ni$^{II}$(PhAc)Br (**4**) and PhICl$_2$ generates an EPR spectrum identical to that of **6** (Supplementary Fig. 39), providing additional experimental evidence for the halide exchange step. Next, we probed the ligand exchange step with an alkoxide. The addition at −80 °C of quinuclidine in MeOH to **5** generates a species assigned as the Ni$^{III}$-aryl alkoxide complex [($^{iPr}$N3) Ni$^{III}$(PhAc)(OMe)]$^{+}$ (**7**), which exhibits a rhombic EPR signal with $g_x = 2.326$, $g_y = 2.151$, $g_z = 2.027$ and superhyperfine coupling to the one axial N ($I = 1$) donor in the $g_z$ direction of A$_z$(N) = 19.4 G (Fig. 5g). We posit that replacement of the halides that exhibit superhyperfine interactions (Br or Cl, $I = 3/2$) by the methoxide ligand resolves the superhyperfine coupling to the N atom along the $g_z$ direction, while the weaker π-donor methoxide

leads to a stronger Ni–N$_{axial}$ interaction. The identity of **7** was further confirmed by the treatment of **5** with 1 equiv NaOMe, which generated an identical EPR spectrum (Fig. 5h and Supplementary Fig. 45). Importantly, to the best of our knowledge this is the first experimental observation of a ligand exchange reaction (either by a halide or alkoxide) occurring at a Ni$^{III}$ metal center, which is typically proposed as a key step in Ni-mediated C–C and C–heteroatom bond formation reactions[55,56]. Finally, we also note that **5** is a catalytically competent species (Supplementary Table 8) and affords the C–O cross-coupled product in 31% yield, suggesting that **5** can successfully enter the catalytic cycle. In this case, the lower yield is likely due to the slight decomposition of **5** in RT during the catalytic reaction.

Upon warming up to room temperature, species **6** and **7** undergo C–O bond-forming reductive elimination, affording 4-methoxyacetophenone together with different amounts of acetophenone (Fig. 6 and Supplementary Figs. 49–54), and consistent with a favorable reductive elimination from a Ni$^{III}$ species[57,58]. Moreover, the addition of excess anionic ligands (Cl$^-$ or OMe$^-$) leads to increased yields of the C–O coupled product (Fig. 6) and faster decomposition of **6** and **7**, as seen in the fast decay of their EPR signals (Supplementary Figs. 42 and 44). We propose that coordination of an additional anionic ligand in the presence of excess base leads to the formation of a congested 6-coordinate Ni$^{III}$ center due to the steric hindrance of the ethylene bridge that blocks the second axial coordination site (Fig. 2d), and consequently results in accelerated C–O reductive elimination. This is also in line with the observed rapid decay of the 6-coordinate species ($^{iPr}$N3)Ni$^{III}$(PhAc)BrCl (**3**, Fig. 4c and Supplementary Fig. 33). In addition, the rapid reductive elimination from a 6-coordinate Ni$^{III}$ center may preclude β-hydride elimination (Fig. 6, entry 6), which is likely the source of the formation of **b**. Overall, these EPR and reactivity studies provide a complete picture of the Ni$^I$/Ni$^{III}$ catalytic cycle for the C–O cross-coupling reaction, with experimental evidence for each essential step in this process: oxidative addition, transmetalation/ligand exchange, and reductive elimination.

**Mechanistic considerations**. Finally, based on the photoredox/Ni dual catalytic studies under blue LED light (Table 2), we evaluated the thermodynamic feasibility of the photocatalytic process in the C–O coupling reaction. A comparison of the redox potentials of the Ni complexes and photocatalysts implies that a reductive quenching pathway is not operating, as the observed production yields contradicts the trend of the oxidation strength of excited photocatalysts (see Supplementary Discussion). Instead, an oxidative quenching pathway is proposed to be operative, in which the initial step is the single-electron reduction of Ni$^{II}$ by the excited photocatalyst, and this is in line with our results in which all photocatalysts with $E(M^+/^*M) < -0.87$ V afford the C–O coupling product. By combining the insights obtained from EPR and reactivity studies and the photoredox/Ni dual catalysis, two photocatalytic mechanisms are proposed (Fig. 7). The ($^R$N3)Ni$^{II}$ complex is reduced to a Ni$^I$ species either by irradiation with purple LED light (Fig. 7a) or via a single

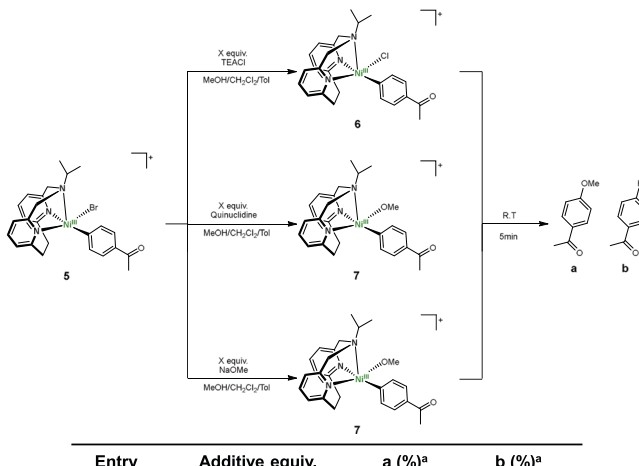

| Entry | Additive equiv. | a (%)$^a$ | b (%)$^a$ |
|-------|-----------------|-----------|-----------|
| 1 | 1 equiv. TEACl | 13 | 20 |
| 2 | 10 equiv. TEACl | 33 | 30 |
| 3 | 1 equiv. Quinuclidine | 28 | 28 |
| 4 | 10 equiv. Quinuclidine | 33 | 22 |
| 5 | 1 equiv. NaOMe | 41 | 24 |
| 6 | 10 equiv. NaOMe | 57 | 0 |

$^a$Yields were determined using GC-FID with 1,3-benzodioxole as a standard. No homocoupled product was formed, and <5% unreacted substrate was observed.

**Fig. 6 Stoichiometric reactivity studies.** Stoichiometric reactivity of 5 with TEACl, quinuclidine, or NaOMe to yield the C–O coupled product through reductive elimination from Ni$^{III}$ intermediates.

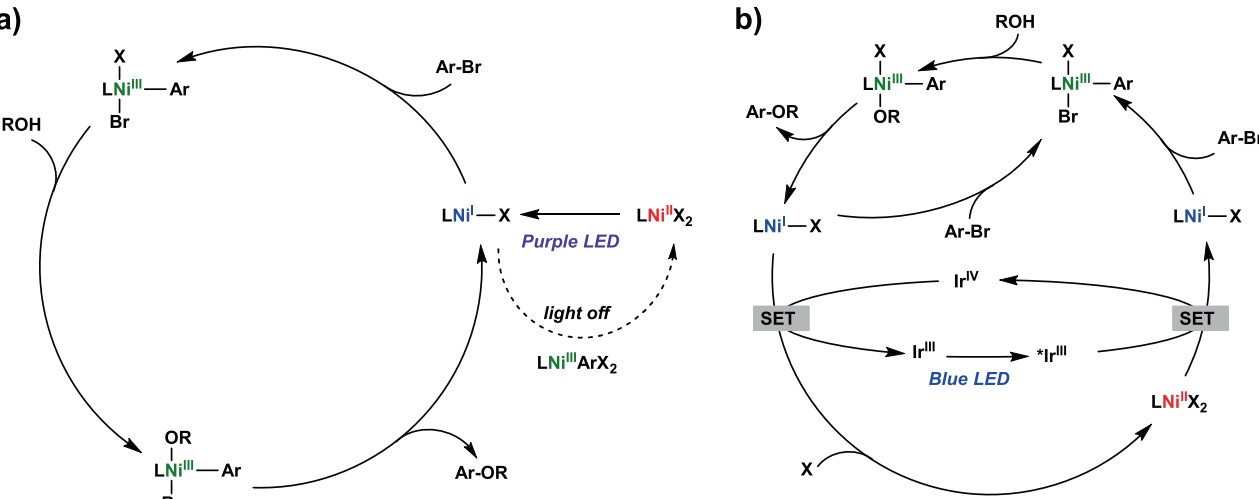

**Fig. 7 Proposed photocatalytic mechanisms.** Proposed mechanisms for the C–O cross coupling reaction for **a** direct light-promoted Ni catalysis with purple LED light (proposed catalyst inactivation pathway in absence of light is shown in dashed arrow), and **b** dual Ni/photoredox catalysis with blue LED light. X = Cl or Br, L = $^{Me}$N3 or $^{iPr}$N3.

electron transfer (SET) from the excited photocatalyst (Fig. 7b), followed by an oxidative addition of the aryl halide to form a transient $(^{R}N3)Ni^{III}Ar(Br)X$ species. For both of the direct light-promoted Ni catalysis and dual Ni/photoredox catalysis, the $(^{R}N3)Ni^{III}$ species can undergo a ligand exchange with the alcohol substrate in presence of excess base to generate the $(^{R}N3)Ni^{III}Ar(OR)X$ species, which is then proposed to undergo a subsequent reductive elimination to yield the C–O coupled product and regenerate a $(^{R}N3)Ni^{I}$ species, completing the catalytic cycle. As the proposed catalytic cycle in Fig. 7a suggests a $Ni^{I}/Ni^{III}$ dark catalytic cycle, then the theoretical quantum yield for the light-promoted Ni catalysis should be above 1 ($\Phi_{Ni} > 1$). The measured quantum yield was determined experimentally to be below 1 ($\Phi_{Ni} < 0.26$, Supplementary information page 32), which is likely due to the less efficient absorption of light in the heterogeneous reaction mixture that contains insoluble material and thus prevents an accurate determination of the number of photons absorbed by Ni under catalytic conditions. In addition, the quantum yield ($\Phi_{Ni} < 0.26$) obtained by actinometry could be significantly overestimated. Given that there are several possible side reactions that would generate off-cycle $Ni^{II}$ species, if $\Phi_{Ni} \ll 1$, then the dark $Ni^{I}/Ni^{III}$ dark catalytic cycle may not be able to compensate for the deleterious side reactions. While values of $\Phi > 1$ were reported previously for similar photoredox/Ni dual catalysis C–O cross-coupling reactions[13], a dark $Ni^{I}/Ni^{III}$ cycle cannot be unambiguously ruled out. To provide further evidence for a dark catalytic cycle, we performed a light on/off experiment and employed ReactIR to monitor in real time the consumption of the starting material 4-bromoacetophenone, which exhibits a characteristic IR C=O stretch at 1690 cm$^{-1}$ (Supplementary information page 34). When the reaction was performed with the light on 25% of the time and the light off 75% of the time, the reaction yield was only marginally lower than when the reaction was performed under continuous illumination (Supplementary Table 9), while the ReactIR monitoring clearly showed the continuous consumption of the substrate during the light-off periods (Supplementary Fig. 25). Overall, these results strongly support a self-sustained $Ni^{I}/Ni^{III}$ cycle, although continuous irradiation during catalysis seems to be beneficial to regenerate any depleted $(^{R}N3)Ni^{I}$ species due to a $Ni^{I}/Ni^{III}$ comproportionation process.

To further probe the proposed $Ni^{I}/Ni^{III}$ comproportionation reaction that can generate off-cycle $Ni^{II}$ species, we investigated this process via EPR and NMR spectroscopy. The reaction between 5 and in situ generated 2 (i.e., 1b + CoCp*$_2$) leads to the rapid decay of the EPR signal, indicating the decomposition of 5 and conversion into an EPR inactive species (Supplementary Fig. 26). In addition, NMR analysis of the reaction confirmed the formation of $(^{iPr}N3)Ni^{II}Cl_2$ complex 1b as the terminal product of comproportionation, along with several organic products (Supplementary Figs. 27–30). These EPR and NMR results thus support the deleterious $Ni^{I}/Ni^{III}$ comproportionation leading to off-cycle $Ni^{II}$-dihalide species, which can re-enter the catalytic cycle upon photoexcitation and is also consistent with previously reported computational[13] and experimental reports[59]. For the proposed dual Ni/photoredox catalysis (Fig. 7b), the $(^{R}N3)Ni^{I}$ species can directly re-enter the catalytic cycle, or it can be oxidized to the $(^{R}N3)Ni^{II}$ complex via an SET process with the oxidized photocatalyst $Ir^{IV}$, thus regenerating the $Ir^{III}$ photocatalyst. Overall, both mechanisms are proposed to involve both $Ni^{I}$ and $Ni^{III}$ intermediates, which are involved in the oxidation addition step, and the ligand exchange/transmetalation and reductive elimination steps, respectively. The main difference between the two pathways is the generation of the catalytically active $(^{R}N3)Ni^{I}$ species, either through light-promoted reduction of the $Ni^{II}$ precursor (Fig. 7a), or by the excited photocatalyst (Fig. 7b).

In conclusion, herein we report the tridentate pyridinophane $^{R}N3$ ligands and their Ni complexes, which were employed in mechanistic studies of the photocatalytic C–O cross-coupling reaction. The $(^{R}N3)NiCl_2$ complexes are active photocatalysts for the C–O cross-coupling in the absence of a precious metal photocatalyst. Employing these developed $^{R}N3$ ligands allowed us to directly probe the key steps of the C–O cross-coupling catalytic cycle involving paramagnetic Ni species: the oxidative addition of an aryl halide to a $Ni^{I}$ species to generate a $Ni^{III}$ species, the ligand exchange/transmetalation step at a $Ni^{III}$ center, and the C–O bond forming reductive elimination from a $Ni^{III}$ species. Overall, the present work suggests that the $^{R}N3$ ligands can lead to the development of improved Ni catalysts, and are also a practical platform for detailed mechanistic studies of related Ni-catalyzed reactions.

## Methods

**General procedure for the C–O cross-coupling reaction.** In a N$_2$-filled glovebox, 4-bromoacetophenone (0.4 mmol, 1.0 equiv), quinuclidine (0.44 mmol, 1.1 equiv), MeOH (1.6 mmol, 4 equiv), $(^{Me}N3)NiCl_2$ (2 mol%), and a magnetic stir bar were added into 0.4 mL THF in a vial. The vial was irradiated with one Kessil purple LED lamp ($\lambda_{max} = 390$ nm, max 52 W) under fan cooling. After 24 h, 1,3-benzo-dioxole (0.4 mmol) was added to the reaction mixture at a standard, and the reaction mixture was analyzed by $^1$H NMR in CDCl$_3$ and GC-FID to determine the yield of C-O coupled product. Authentic 4-methoxyacetophenone was purchased from AK scientific and used to determine the retention time and response factor for GC-FID quantification.

**General procedure for EPR detection of 2 and 3.** In a N$_2$-filled glovebox, a MeOH solution of 1b was layered on top of a frozen PrCN solution of CoCp*$_2$ and 4-bromoacetophenone in the EPR tube. The mixture was quickly frozen and taken outside of the glove box. The reaction mixture was mixed at −95 °C, quickly frozen at 77 K, and then the sample was warmed up for ~10 s to −95 °C to allow for a complete reaction.

**General procedure for EPR detection of 5, 6, and 7.** In a N$_2$-filled glovebox, an EPR tube was charged with 1:3 CH$_2$Cl$_2$:Toluene or CH$_2$Cl$_2$:PrCN solution of 5 and frozen at 77 K. A MeOH solution of TEACl, quinuclidine, or NaOMe was added over the frozen solution of 5. After taking out the EPR tube from the glovebox, an initial EPR spectrum was taken at 77 K (for 5). After a quick shake of the tube for 10 s at −95 °C (for 6 or 7), the sample was frozen at 77 K and the EPR spectrum was collected.

## Data availability

The crystallographic data generated in this study have been deposited in the Cambridge Crystallographic Data Center database under deposition numbers CCDC 2085516 (1a) and 2085517 (1b). The synthetic details, spectroscopic and electrochemical characterization of new compounds, computational details, mechanistic studies, and X-ray crystallographic data generated in this study are provided in the Supplementary Information file. Any additional data that support the findings of this study are available from the corresponding author upon request.

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

## Acknowledgements

We thank National Science Foundation (CHE-1925751) for financial support. We also thank Leonel Griego for ReactIR experiments and helpful discussions, and Dr. Richard T. Haasch for the XPS analysis.

## Author contributions

L.M.M. conceived the initial ligand synthesis procedure. H.N. and L.M.M. conceived and designed the experiments, H.N. carried out the experimental work, L.M.M. performed computational calculations, and H.N. analyzed the computational and experimental data. H.N. and L.M.M. wrote the manuscript.

## Competing interests

A provisional patent (U.S. Patent Application No. 63/215,131, Title: "Tridentate Macrocyclic Compounds", Inventors: Liviu M. Mirica and Hanah Na, Filing date: Jun 25, 2021) has been filed on the synthesis and applications of the tridentate pyridinophane ligands. All authors declare no other competing interests.
