## [Peer Review File · Nature Communications]

REVIEWER COMMENTS

Reviewer #1 (Remarks to the Author):

The Authors report the synthesis and characterization of a tridentate pyridinophane ligand and the Ni(I), and Ni(II), and Ni(III) complexes derived therefrom. The Authors also demonstrate the relevance of the resulting Ni complexes in the context of a photoredox C-O cross-coupling reaction by showing that they may serve as intermediates towards product formation. Although the intermediacy of Ni(I), Ni(II), and Ni(III) states currently serve as the dominant mechanistic paradigm in photoredox-mediated Ni-catalyzed cross-coupling reactions, this work represents the first example where a catalytically-relevant Ni(III) species often proposed to be necessary for reactivity has been isolated and characterized. As such, this manuscript reports an 'missing link' in the mechanistic study of photoredox cross-coupling reactions, and I feel it will be of interest to the readership of Nature Communications. However, there are a few points which I hope that the authors can address or clarify before this manuscript is accepted for publication:

1) The UV-vis spectrum shown in Fig 3 is only of the Ni(II) complex by itself. However, under the photoredox conditions, quinuclidine and MeOH can both serve as competent ligands and possibly change the UV-vis features upon coordination. The authors should report the UV-vis spectrum of the Ni(II) compounds under the complete reaction conditions since the lack of reactivity under blue LED irradiation in the absence of an additional photocatalyst is justified by the absorption spectrum of Ni(II).

2) The Authors justified the presence of a dark cycle in the cross-coupling reaction by a comparison between entries 2 and 3 in Table S9, which showed that a sample irradiated for a total of 5 h with intervals where it was stirred in the dark showed a higher yield than one subjected to 5 h of continuous irradiation. Were these measurements performed just once or in triplicate? Furthermore, would it be possible to measure the quantum yield of the reaction in order to provide further evidence for a dark cycle?

3) In the stoichiometric reactions shown in Figure 6, which presumably go through the same intermediates as the full photoredox reaction, the yield of Ar-OMe is at most 57% while many of the conditions also give significant amounts of arene side product. This seems problematic since the catalytic reactions can achieve yields in excess of 90% without significant arene formation. Could the Authors provide a rationale for why this may be? Also, in Entry 6, was the balance of the reaction just unreacted starting material or were other side products observed?

4) In Figure 7 the Authors propose oxidative quenching of the Ir(III) photocatalyst as the means by which Ni(II) is reduced down to Ni(I) to enter the cycle. Another possibility which has been previously reported is that quinuclidine acts as a flash-quencher of Ir(III)* in order to generate Ir(II), the latter of which reduces the Ni(II) species. Could the Authors address this latter pathway for Ni reduction and discuss why they favour the oxidative quenching mechanism?

**

Editorial note: In comments to the editor, Reviewer 1 highlighted that the EPR spectrum for 2 + 3 in a 0.6:1 ratio (Figure 4e) does not match the simulation. The reviewer asks for an explanation, and if there is a simulation ratio that better fits the data. As the assignment of the structure of 3 seems to be model-dependent, we recommend either strengthening the evidence for the intermediate, or softening the language around the assignment throughout the manuscript.

Reviewer #2 (Remarks to the Author):

This paper by Na and Mirica describes the synthesis, characterization, and detailed reactivity studies of isolated Ni(I) and Ni(III) pyridinophane complexes. These complexes represent stabilized models of important intermediates proposed for Ni-catalyzed etherification reactions of aryl halides. Such intermediates are often proposed but seldom, if ever, detected in catalysis. The present work therefore fills in significant gaps our community's working knowledge of these reactions even if the results are largely as originally proposed. The experiments are overall well executed and bring important spectroscopic characterization of intermediates relevant to this specific area of organometallic catalysis. I recommend it for publication in Nature Communications once the following points are addressed.

- A major portion of the manuscript is spent trying to sell the novelty of the N-3 pyridinophane ligand, which, while interesting, is not well differentiated from the others mentioned in their own introduction. The authors state other N3 ligands "were shown to only stabilize high-valent Ni species" yet it is not clear if anyone tried (a formal Ni(I) Tp complex HAS been reported: Dalton Trans., 2016, 45, 14581). It's not immediately evident to me why this ligand has the unique ability to stabilize both Ni(I) and Ni(III) relative to other N3 ligands. This is important to note because the authors highlight some rather impressive catalysis results with their ligand and some of the other N3 ligands are much easier to make. The community would be far more interested to know if Tp, Tpm, Tp*, Tpm*, and TACN are as active in catalysis as some of these are commercial. Overall it feels like the authors are trying to sell this ligand as more than the data support.

I would recommend that the authors change their language in the justification of this ligand and just leave it at a N3 derivative of the N4 ligand that is a cornerstone of their research program and that this will stabilize Ni(I) better. The introduction reads as if this ligand is expected to be significantly better than other N3 ligands. It might be, but the data are insufficient to say that "sterics promote reductive elimination" if it hasn't been shown that other Ni(III)(OR)(Ar) complexes struggle to mediate C-O bond formation, among other claims.

- I think that the catalysis results are significant. I would recommend the authors try some of the catalysis with other N3 ligands. I don't think it is necessary for publication but do think people would like to know of Tpm, Tp*, Tpm*, and TACN derivatives can catalyze these transformations as efficiently as their ligand.

- I think that some of the key cyclic voltammograms should be added to the body of the text and discussed further. There are some interesting features that I feel are relevant to catalysis and broader mechanistic conclusions. For example, the reduction of the 1a and 1b are highly irreversible. To me this is indicative of a significant chemical structural change beyond just coordination of a pendant donor and loss of Cl. Certainly we don't expect a 59mv peak separation on the return oxidation, however, the total absence of oxidation in the cathodic sweep suggests that there is something more going on. This is important because the presumed speciation Ni(I) product is a basis of the paper. It might also relate to some unexpected observations in figures 4 and 5, namely the unusual stability of 5 relative to 3. The authors should discuss potential reasons for this behavior further instead of burying it in the SI.

- Related to the above redox behavior, others have evidence that dimerization of Ni(I) species is an important consideration in this sort of catalysis and that such dimers do affect activity. If this N3 ligand shuts down this behavior, it would significantly improve the manuscript to know. I would expect this behavior to be attenuated with N3 relative to N2 ligands and I don't think this is well articulated in the literature and could help explain their interesting catalysis results (e.g. the high activity and relatively high concentration of catalysis)

- The relevance of Ni(I)/(III) comproportionation to Ni(II) needs further discussion. Does 1b + CoCp2* (s20) decompose into 4? If so, how does this relate to sustained Ni(I)/(III) catalysis. The low/modest yields in reductive elimination studies in figure 6 mirror previous RE studies from Ni(III) and suggest that comproportionation may limit sustained Ni(I)/(III) catalysis (Nocera, Ref 14). However, figure 4b might suggest otherwise. Overall, I think the present work is highly relevant to previous reports of sustained Ni(I)/(III) catalysis but is not suitably related

to these studies. The impact of the work would be more clear if these details were clarified through experiment and explanation.

•The NMRs in figure S1 don't appear to be appropriately referenced

Reviewer #3 (Remarks to the Author):

In this paper, the authors have developed a new class of tridentate pyridinophane ligands (RN3), with two specific examples of R= H and Me, which when formed into Ni containing complexes act of photocatalysts for C-O coupling reactions. Importantly, the present ligands permit the investigation of the key steps of the catalytic cycle. The Ni complexes are characterized by X-ray diffraction, cyclic voltammetry (CV), ¹³C and ¹H NMR, and UV-vis absorption. The CV nicely illustrates the possibility of the ligand stabilizing both NiI and NiIII species. The authors then study the photocatalytic C-O coupling mediated by the (RN3)NiCl₂ catalysts using the reaction of methanol and 4-bromoacetophenone as an example. The authors carry out useful control experiments, and explore conditions for reaction optimization. While these controls and optimizations are important, the more interesting part of the work lies in the reactivity studies, with the present ligands, related complexes, slightly different reaction conditions, or alternate photocatalysts, and their probing with EPR. For example, from their EPR spectra in fig. 4, they suggest that they have directly observed "the oxidation addition of an aryl halide to NiI species to generate a NiIII species" (see comment 7 below). Overall the experimental work appears to be outstanding and the insight obtained would be of great interest to the community. However, my primary expertise is in computational chemistry, and there are some issues outlined below in the present work. If these can be satisfactorily addressed, and based on the the experimental work, I could tentatively recommend publication.

1. The authors state that their computations have used the M06-tzvp functional. There is no such density functional and I assume the authors mean M06-2X (or M06, or M06-HF or M06-L, or ...). The name of the functional used should be corrected and the authors should include a reference in Section 14 of the SI to the original paper where the functional was first introduced. My rest of my comments are based on the assumption that the M06-2X functional was used in the present work.

2. The authors state that they have used the modified 6-31G* (m6-31G*) basis set for nickel. This basis set is not available from the basis set library in Gaussian; therefore, from where did the authors obtain it. They should also reference the original work for this basis set, i.e., The Journal of Chemical Physics 118(17):7775-7782.

3. The authors justify their choice of functional and basis set by stating "this combination of hybrid functional and basis sets has been previously shown to work well for reproducing experimental parameters for Ni complexes.^{15,16}" Both these references use the B3LYP functional not the one used in the present work, and hence, they cannot be used as justification for the choice used here. Please note that I am not advocating for the use of B3LYP, there are probably other better functionals for the properties (i.e, excited states) considered here, see comment 5.

4. The authors state that "the ground state of 1a has triplet spin multiplicity" but no mention is made in the Computational section about defining the spin of the complexes. I assume that this was done, and they were defined as triplets, and thus the question arises as to whether the DFT computations were carried out using unrestricted (default in Gaussian) or restricted formalism. This choice needs to be clearly defined. Moreover, there are significant challenges is using TD-DFT to compute triplet excited states from an unrestricted triplet ground state; in particular, the states computed can be plagued with spin-contamination. Have the authors examined the spin purity of the resultant excited states, especially the two excited states of primary interest?

5. The authors have assigned the two primary excitations to CT excitations (at 380 nm) and a metal-centred transition (at 640 nm). There are important considerations about the choice of functional for CT transitions; for many hybrid functionals, their use leads to artificially low

excitation energies for CT states, and thus energetic ordering is incorrect, see e.g, J Chem Phys. 2008 Jan 28;128(4):044118. I wonder about the extremely low-energy states (< 1 eV) seen in the long list of computed excited states, see Table S19 and S23 where 100 excited triplet (??) states are reported; note, it is extremely unlikely for TD-DFT to capture this many states correctly although it may provide a reasonable representation for the important low-lying bright states. These CT states can be accounted for by using a long-range corrected functional, e.g, ω B97xD, CAM-B3LYP. It would be useful for the authors to confirm their state assignments using one of these functionals.

6. On page 8, the authors write that "the MC(d-d) state is the lowest excited state" but their TD-DFT results (Tables S19 and S23) show 4 lower-lying excited states, see also comment 5. The authors also refer to "the higher energy CT state and MC(d-d) state are close in energy, thus resulting in fast relaxation to the lowest lying MC(d-d) state." However, these states differ in energy by greater than 1 eV; this is not close so does not seem to support the statement.

7. The authors propose the presence of a 6-coordinate NiIII species (3) on the basis of EPR measurements. Have the authors considered computing the EPR g-tensor to try and confirm the presence of this species?

Minor points

8. For the simulated UV-vis spectra, the line shape and width for each of the contributing peaks should be provided.

REVIEWER COMMENTS

Reviewer #1 (Remarks to the Author):

The Authors report the synthesis and characterization of a tridentate pyridinophane ligand and the Ni(I), and Ni(II), and Ni(III) complexes derived therefrom. The Authors also demonstrate the relevance of the resulting Ni complexes in the context of a photoredox C-O cross-coupling reaction by showing that they may serve as intermediates towards product formation. Although the intermediacy of Ni(I), Ni(II), and Ni(III) states currently serve as the dominant mechanistic paradigm in photoredox-mediated Ni-catalyzed

cross-coupling reactions, this work represents the first example where a catalytically-relevant Ni(III) species often proposed to be necessary for reactivity has been isolated and characterized. As such, this manuscript reports an 'missing link' in the mechanistic study of photoredox cross-coupling reactions, and I feel it will be of interest to the readership of Nature Communications. However, there are a few points which I hope that the authors can address or clarify before this manuscript is accepted for publication:

1) The UV-vis spectrum shown in Fig 3 is only of the Ni(II) complex by itself. However, under the photoredox conditions, quinuclidine and MeOH can both serve as competent ligands and possibly change the UV-vis features upon coordination. The authors should report the UV-vis spectrum of the Ni(II) compounds under the complete reaction conditions since the lack of reactivity under blue LED irradiation in the absence of an additional photocatalyst is justified by the absorption spectrum of Ni(II).

Our response: *We thank the reviewer for this comment. We examined the UV-vis absorption spectra of the reaction mixture, Ni complex+quinuclidine+4-bromoacetophenone+MeOH (a molar ratio of Ni:Ar-Br:quinuclidine:MeOH = 1:50:55:200 was maintained, corresponding to the ratio in the photocatalytic reaction). The enhancement in absorption <400 nm originates from the strong absorption bands of quinuclidine and 4-bromoacetophenone, as seen in the obtained absorption spectra without the Ni complex. In addition, we do not observe any significant change in the absorption bands position or absorptivity in the >400 nm region. We postulate that the lack of reactivity under blue LED irradiation could be due to the very weak absorption in the 440–495 nm range (corresponding to the blue light region), and thus resulting in inefficient photoexcitation. This is in line with the TD-DFT calculation results, where no oscillator strength around 440 nm was predicted, indicating that the probability of excitation with 440 nm light is very small. We have included these UV-vis absorption spectra in the Supplementary Information page S24.*

2) The Authors justified the presence of a dark cycle in the cross-coupling reaction by a comparison between entries 2 and 3 in Table S9, which showed that a sample irradiated for a total of 5 h with intervals where it was stirred in the dark showed a higher yield than one subjected to 5 h of continuous irradiation. Were these measurements performed just once or in triplicate? Furthermore, would it be possible to measure the quantum yield of the reaction in order to provide further evidence for a dark cycle?

Our response: *We thank the reviewer for this comment. We performed two more independent experiments on different instances and have reported the error of the measurements in Supplementary Information page S29. Overall, the on/off light experimental results still hold and support a potential dark catalytic cycle.*

For photochemical quantum yield (QY) measurement, we examined the photon flux of purple LED setup using Potassium ferrioxalate chemical actinometry. The photon flux determined 3.274×10^{-7} einsteins sec^{-1} sample^{-1} . The quantum yield of the photochemical reaction can be determined by the following equation:

$$\Phi = \frac{\text{Rate of product formation (moles/sec)}}{\text{Total photon flux (moles/sec)} \times (1 - 10^{-A(390\text{nm})})}$$

Fraction factor $f = 1 - 10^{-A(390\text{nm})}$, $A(390)$ = absorbance at 390 nm of Ni complex in the solution, which is calculated from the absorption coefficient and concentration ($A(390\text{nm}) = \epsilon_{390\text{nm}} \times \text{concentration}$)

of Ni complex) Considering the low concentration of Ni complex in the reaction mixture (due to the limited solubility in THF), $f = 0.4087$ was used.

Therefore

$$\Phi = \frac{\text{Rate of product formation (moles/sec)}}{(3.274 \times 10^{-7}) \times (0.4087)} = \frac{\text{Rate of product formation (moles/sec)}}{1.338 \times 10^{-7}}$$

It means that for $QY = 1$ (100%), 1.338×10^{-7} moles of the product should be obtained per second, indicating a fast reaction rate. When the reaction mixture of 0.8 mL (0.8 mmol of 4-bromoacetophenone is used) is irradiated, the C–O coupled product needs to be obtained 100% in 5979 sec = 1.66 hr, theoretically.

However, in our reaction condition, completion of the reaction takes longer than 5 h, indicating a $QY < 1$. Considering the average 78% yield obtained following the 5 h irradiation in light on/off experiments, the calculated QY is 0.26 (26%), while the average 60% yield obtained for the continuous 5 h irradiation resulted in the QY of 0.19 (19%).

We postulate that the lower QY is likely due to the less efficient absorption of light in the heterogeneous reaction mixture that contains insoluble material and could prevent the accurate determination of the number of photons absorbed by Ni in the catalytic condition. In addition, it has been known that the calculated QY of the reaction could vary during the reaction, indicating reaction rate changes. For example, Knowles et al., in Chem. Sci. 2016, 7, 2066, showed that depending on when the reaction mixture was analyzed to calculate [mole of product/time], the QY changes from 0.13 to 2, and this does not necessarily represent a change in mechanism, but rather reflects changes in the reaction rate. The QY results are included in the Supplementary Information page S31.

ReactIR reaction monitoring: To provide further evidence for a dark catalytic cycle, we performed a light on/off experiment and ReactIR to monitor in real time the consumption of the starting material 4-bromoacetophenone, which exhibits a characteristic IR C=O stretch at 1690 cm^{-1} (Supplementary Information, pages S34). When the reaction was performed with the light on 25% of the time and 75% of the time with light off, the reaction yield was only marginally lower than when the reaction was performed under continuous illumination (Supplementary Information, Table S9), while the ReactIR monitoring clearly showed the continuous consumption of the substrate during the light-off periods (Supplementary Information, Fig. S24).

As shown in Figure S24, in the 30 min light on/5 min light off experiments a continuous consumption of substrate was observed in the corrected trend line (green solid line), including the light off period. For the 20 min light on/20 min light off experiments the corrected trend line was noisier, and then the slopes corresponding to the substrate consumption between the reaction with and without **1a** during the light-off time were compared. Reaction with **1a** exhibits a noticeable decrease of the signal intensity during light off-time (solid red line, black dash), suggesting the consumption of the substrate. In contrast, in the absence of **1a** there is a negligible signal intensity decrease (solid blue line, black dash). Taken together, these results support a self-sustained $\text{Ni}^{\text{I}}/\text{Ni}^{\text{III}}$ dark cycle, although throughout the catalysis a continuous irradiation is likely needed to regenerate the depleted active Ni^{I} species due to the $\text{Ni}^{\text{I}}/\text{Ni}^{\text{III}}$ comproportionation process (see page S37 for further discussion).

Figure S24. ReactIR reaction progress trend lines for 4-bromoacetophenone consumption (at 1690 cm^{-1}) on an enlarged scale with light 30 min on/5 min off (left), and 20 min on/20 min off (right). To allow for a clear visualization of all trend lines in one figure, some trend lines were moved up or down on the Y axis by an arbitrary value. Red solid line: reaction with 1a. Blue solid line: reaction without 1a.; green solid line: corrected trendline, showing a continuous consumption of the substrate.

While values of $\Phi > 1$ were reported previously for similar photoredox/Ni dual catalysis C – O cross-coupling reactions, a dark $\text{Ni}^{\text{I}}/\text{Ni}^{\text{III}}$ cycle seems to be operative since higher yields were obtained in the light on/off experiments than the comparable light-on experiments (Table S9), and ReactIR studies suggest substrate consumption during the light off periods. This is in line with previous reports on similar photoredox/Ni dual catalysis for C–O cross-coupling reactions (Nocera et al, *J. Am. Chem. Soc.* 2019, 141, 89). We have included ReactIR experiment details and discussion in the main text (page 21–22) and supporting information (Supplementary Information page S34), respectively.

3) In the stoichiometric reactions shown in Figure 6, which presumably go through the same intermediates as the full photoredox reaction, the yield of Ar-OMe is at most 57% while many of the conditions also give significant amounts of arene side product. This seems problematic since the catalytic reactions can achieve yields in excess of 90% without significant arene formation. Could the Authors provide a rationale for why this may be? Also, in Entry 6, was the balance of the reaction just unreacted starting material or were other side products observed?

Our response: We thank the reviewer for this comment. We assume that the reason for the significant arene formation might be found in the difference between stoichiometric reaction conditions and catalytic conditions. Considering that adventitious water present in the solvents may induce protodehalogenation, the amount of MeOH might be one of the reasons. In the catalytic condition, a ratio of MeOH:organic solvent=1:6 (v:v) was used, while in the stoichiometric conditions a ratio of MeOH:organic solvent=1:2 or 3 (v:v) was used in order to dissolve a decent amount of TEACl or NaOMe, and thus an increased amount of adventitious water may be present. This is in line with the observation that in presence a larger amount of strong base, less acetophenone was observed (entry 3 vs 4 and entry 5 vs 6). In addition, currently it is not easy to determine whether acetophenone is originating from complex 5 (before reacting with additives) or complex 6 or 7. Although complex 5 is stable, there is a chance that decomposition

could occur before reacting with additives, during the synthesis or handling (preparing solution, transferring solution etc).

In entry 6, we do not observe any other side product, such as homo-coupled 4,4'-diacetylbiphenyl in GC-MS analysis. We assume that the reaction (either ligand exchange or reductive elimination) was not complete, yet the reason for the incomplete reaction is not apparent yet. During these experiments, we found that the color of the reaction solution in Entry 6 stays yellow-orange even after 5 min at RT, suggesting the presence of unreacted/non-decomposed Ni complex. Therefore, we performed the same reaction, yet acid workup was carried out before GC-MS analysis to decompose any remaining Ni complex, and in this case we do observe both acetophenone and 4-methoxyacetophenone. Moreover, please note that trace amount (< 5%) of 4-bromoacetophenone was observed in all cases (Figures S48–53), corresponding to the small peak with a retention time of 9.3 min, before the peak for 4-methoxyacetophenone with a retention time of 9.5 min. We have included the footnote in the main text table (page 20, Figure 6) for clarification: “No homocoupled product was formed, and <5% unreacted substrate was observed.”

4) In Figure 7 the Authors propose oxidative quenching of the Ir(III) photocatalyst as the means by which Ni(II) is reduced down to Ni(I) to enter the cycle. Another possibility which has been previously reported is that quinuclidine acts as a flash-quencher of Ir(III)* in order to generate Ir(II), the latter of which reduces the Ni(II) species. Could the Authors address this latter pathway for Ni reduction and discuss why they favour the oxidative quenching mechanism?

Our response: We thank the reviewer for this comment. The reviewer makes a good point that quinuclidine could be oxidized to quinuclidine radical cation by reducing $*Ir^{III}$ to Ir^{II} , depending on the $E(*Ir^{III}/Ir^{II})$ of the photocatalyst. Considering $E(Q^+/Q) = +1.09$ V, the reaction between quinuclidine and excited photocatalysts will only occur when $E(*M/M^+)$ of photocatalyst is larger than +1.09 V, meaning that it can happen with two of photocatalysts that examined in our study, $[Ru(bpz)_3](PF_6)_2$ and $[Ir(dF(CF_3)_2ppy)(dtbbpy)]PF_6$. While $[Ir(dF(CF_3)_2ppy)(dtbbpy)]PF_6$ successfully generated the C–O product, $[Ru(bpz)_3](PF_6)_2$ failed to give C–O coupling product. Although the failure of $[Ru(bpz)_3](PF_6)_2$ to give C–O product could be due to an insufficient thermodynamic driving force to reduce Ni^{II} to Ni^I ($E(Ru^{II}/Ru^I) = -0.80$ V and $E(Ni^{II}/Ni^I) = 0.82$ V are very close in value), we could observe the C–O product formation with *fac*- $Ir(ppy)_3$ and $[Ir(ppy)_2(dtbbpy)]PF_6$, (with $E(*Ir^{III}/Ir^{II}) = +0.31$ V and 0.66 V, respectively), for which $*Ir^{III}$ cannot be quenched by quinuclidine. Therefore, we consider that the reductive quenching cycle employing quinuclidine is less likely to be operating during this catalytic reaction. We have included this discussion on the quinuclidine-quenching possibility in the Supplementary Information page S43 (section 8; Thermodynamic feasibility examination of photocatalytic process).

**

Editorial note: In comments to the editor, Reviewer 1 highlighted that the EPR spectrum for 2 + 3 in a 0.6:1 ratio (Figure 4e) does not match the simulation. The reviewer asks for an explanation, and if there is a simulation ratio that better fits the data. As the assignment of the structure of 3 seems to be model-dependent, we recommend either strengthening the evidence for the intermediate, or softening the language around the assignment throughout the manuscript.

Our response: We thank the reviewer for this comment. We checked our simulation and also examined other ratios, as shown in the following figure. We tried 2:3 = 0.5:1 ratio ((EPR spectrum of 2)×0.5 + EPR spectrum of 3), 0.6:1 ratio ((EPR spectrum of 2)×0.6 + EPR spectrum of 3), 0.7:1 ratio ((EPR spectrum of 2)×0.7 + EPR spectrum of 3), 1:1 (EPR spectrum of 2 + EPR spectrum of 3). Considering the second and third peaks (marked with *), we believe that linear addition of the EPR spectra of 2 and 3 in a 0.6:1 ratio, both for the experimental EPR spectra (left) or simulated EPR spectra (right), gives the best fits to the obtained experimental EPR spectrum in Figure 4e. We have also softened the language for the tentative assignment of species 3 to be labeled as “proposed” or “putative” throughout the text. Also, please keep in mind that we specifically labeled the structure for species 3 as ‘proposed’ in Fig. 4!

(Left) An examination of various ratios of 2:3 employed in simulating the experimental EPR spectrum in Figure 4e using experimental EPR spectra of 2 and 3. (Right) An examination of ratio of 2:3 in experimentally obtained EPR spectrum in Figure 4e, using simulated EPR spectra of 2 and 3.

Reviewer #2 (Remarks to the Author):

This paper by Na and Mirica describes the synthesis, characterization, and detailed reactivity studies of isolated Ni(I) and Ni(III) pyridinophane complexes. These complexes represent stabilized models of important intermediates proposed for Ni-catalyzed etherification reactions of aryl halides. Such intermediates are often proposed but seldom, if ever, detected in catalysis. The present work therefore fills in significant gaps our community’s working knowledge of these reactions even if the results are largely as originally proposed. The experiments are overall well executed and bring important spectroscopic

characterization of intermediates relevant to this specific area of organometallic catalysis. I recommend it for publication in Nature Communications once the following points are addressed.

•A major portion of the manuscript is spent trying to sell the novelty of the N-3 pyridinophane ligand, which, while interesting, is not well differentiated from the others mentioned in their own introduction. The authors state other N3 ligands “ were shown to only stabilize high-valent Ni species” yet it is not clear if anyone tried (a formal Ni(I) Tp complex HAS been reported: Dalton Trans., 2016, 45, 14581). It’s not immediately evident to me why this ligand has the unique ability to stabilize both Ni(I) and Ni(III) relative to other N3 ligands. This is important to note because the authors highlight some rather impressive catalysis results with their ligand and some of the other N3 ligands are much easier to make. The community would be far more interested to know if Tp, Tpm , Tp*, Tpm*, and TACN are as active in catalysis as some of these are commercial. Overall it feels like the authors are trying to sell this ligand as more than the data support. I would recommend that the authors change their language in the justification of this ligand and just leave it at a N3 derivative of the N4 ligand that is a cornerstone of their research program and that this will stabilize Ni(I) better. The introduction reads as if this ligand is expected to be significantly better than other N3 ligands. It might be, but the data are insufficient to say that “sterics promote reductive elimination” if it hasn’t been shown that other Ni(III)(OR)(Ar) complexes struggle to mediate C-O bond formation, among other claims.

Our response: *We thank the reviewer for these comments. We apologize for misleading the reader, our intention for mentioning other tridentate ligands was mainly to highlight the difference between those and our ^RN3 ligand (and why we wanted to develop a new ^RN3 ligand, although there are other known tridentate ligands), and our intention was not to imply that the new ^RN3 ligand is superior to them.*

Regarding the comparison of newly developed ^RN3 ligands and other known tridentate ligands in the introduction, we missed including a critical explanation in the original manuscript. It has been reported that for some of the known tridentate ligands, monomeric Ni dihalide species (such as a (Ligand)NiX₂ complex) cannot easily be obtained. For example, the reaction of simple Me₃TACN ligand and NiCl₂ source gave cationic trichloride Ni dimeric structure, [(Me₃TACN)₂Ni₂(μ-Cl)₃]⁺ (Inorg. Chem. 1997, 36, 2834). In addition, it has been reported that the reactions of NiX₂ with trispyrazolylmethane (Tpm), tris(3,5-dimethylpyrazolyl)methane (Tpm), or hydrotris(3,5-dimethylpyrazolyl)borate (Tp*) readily generate homoleptic 2:1 L:Ni complexes [L₂Ni]²⁺ (L = Tpm, Tpm*, or Tp*) as the thermodynamic product, instead of LNiX₂ (Inorganica Chimica Acta 2006, 359, 2592). There was an attempt to synthesize Tpm and Tpm* NiX₂ species, using steric constraints on the ligand to prevent the formation of the homoleptic L₂Ni species (Inorg. Chem. 2009, 48, 3535). Although the work successfully found the synthetic route for single Tpm-ligated Ni halide complex, the obtained crystal structure includes coordination of H₂O, and more importantly it was mentioned that “TpmNiX₂(H₂O)_n are kinetic products, and although they are stable indefinitely in the solid-state, they readily convert to the thermodynamic product [(Tpm)₂Ni]²⁺ in solution over the course of several hours at room temperature”. In contrast, a simple reaction of ^RN3 ligand and NiCl₂ source exclusively generates (^RN3)NiX₂, which we consider is an advantage of our ^RN3 ligands. We have included an expanded explanation of the difference between the ^RN3 ligands and other tridentate ligands in the revised manuscript pages 3 and 6.*

Taken all together, we believe that new pyridinophane-based ^RN3 ligands are more suitable for our targeted study of providing mechanistic evidence for the reactivity mimicking (dtbbpy)NiCl₂ systems, which cannot be easily achieved with other Tp, Tpm, or Me₃TACN tridentate ligands, and thus it highlights

the unique characteristic of the newly developed pyridinophane based $^R\text{N}_3$ ligand. We hope our comments clarify the difference between new $^R\text{N}_3$ ligand and other known tridentate ligands.

We also thank the reviewer for mentioning the reference reporting a mononuclear $\text{TpmNi}^{\text{I}}\text{-CO}$ complex, we have included that reference in the revised manuscript.

Finally, we clearly agree with the reviewer's viewpoint that it is not appropriate to claim that our $^R\text{N}_3$ ligand is unique for stabilizing Ni^{I} and Ni^{III} species compared to other tridentate N-donor ligands. In addition, as the reviewer commented, reductive elimination of other $\text{LNi}^{\text{III}}(\text{OR})(\text{Ar})$ complex has not been examined, thus we agree it is inappropriate to say that "steric promotes reductive elimination". Therefore, we have eliminated the sentences "yet these systems were shown to only stabilize high-valent Ni species" and "Importantly, the use of tridentate pyridinophane $^R\text{N}_3$ ligands generates a sterically hindered 6-coordinate Ni^{III} center due to the H atoms of the ethylene bridge and that is proposed to promote a rapid reductive elimination."

•I think that the catalysis results are significant. I would recommend the authors try some of the catalysis with other N_3 ligands. I don't think it is necessary for publication but do think people would like to know of Tpm , Tp^* , Tpm^* , and TACN derivatives can catalyze these transformations as efficiently as their ligand.

Our response: We thank the reviewer for this comment. We have performed the direct light-promoted catalysis with a Ni complex supported by another tridentate ligand, $(i\text{Pr}_3\text{TACN})\text{NiCl}_2$, and have obtained a 51% yield of C–O cross-coupled product. The result has been included in the Supplementary Information, table S8 (entry 7).

•I think that some of the key cyclic voltammograms should be added to the body of the text and discussed further. There are some interesting features that I feel are relevant to catalysis and broader mechanistic conclusions. For example, the reduction of the 1a and 1b are highly irreversible. To me this is indicative of a significant chemical structural change beyond just coordination of a pendant donor and loss of Cl. Certainly we don't expect a 59mv peak separation on the return oxidation, however, the total absence of oxidation in the cathodic sweep suggests that there is something more going on. This is important because the presumed speciation $\text{Ni}(\text{I})$ product is a basis of the paper. It might also relate to some unexpected observations in figures 4 and 5, namely the unusual stability of 5 relative to 3. The authors should discuss potential reasons for this behavior further instead of burying it in the SI.

Our response: We thank the reviewer for this comment. We have included the cyclic voltammograms of **1a** in Figure 3c. In addition, as the reviewer suggested, we have included explanations for the irreversible reduction waves, which are related to the key step for entering the catalytic cycle, i.e. reduction of Ni^{II} to Ni^{I} species. The first reduction wave is proposed to correspond to the conformation in which the $^R\text{N}_3$ ligand adopts a κ^2 binding mode, in line with the flexibility of the pyridinophane ligands observed by us previously (Mirica et al, *Inorg. Chem.* 2014, 53, 13112 and *J. Am. Chem. Soc.* 2017, 139, 35). The electrochemical irreversibility of the following reduction wave is likely due to the loss of a Cl ligand and the change in coordination geometry from square pyramidal to tetrahedral upon formation of Ni^{I} species (i.e. an EC mechanism: electron transfer followed by chemical reaction EC mechanism, see Zanello, P. *Inorganic Electrochemistry. Theory, Practice and Application*; RSC: Cambridge, U.K., 2003).

•Related to the above redox behavior, others have evidence that dimerization of Ni(I) species is an important consideration in this sort of catalysis and that such dimers do affect activity. If this N3 ligand shuts down this behavior, it would significantly improve the manuscript to know. I would expect this behavior to be attenuated with N3 relative to N2 ligands and I don't think this is well articulated in the literature and could help explain their interesting catalysis results (e.g. the high activity and relatively high concentration of catalysis)

Our response: We thank the reviewer for this comment. As the reviewer pointed out, the $(^R\text{N}3)\text{NiCl}_2$ catalyst exhibits activity at higher concentrations and lower catalyst loadings (<2 mol% Ni loading, 1 M concentration) vs. the several other known C–O coupling reactions employing bidentate ligands such as the $(^t\text{Bu}bpy)\text{Ni}$ systems, e.g. Xu et al, *Angew. Chem. Int. Ed.*, 2020, 59, 12714; $(^t\text{Bu}bpy)\text{NiArBr}$ 5mol%, 0.5 M (increasing concentration to 1 M was detrimental), Nocera et al, *Angew. Chem. Int. Ed.*, 2020, 59, 9527; $(^t\text{Bu}bpy)\text{NiCl}_2$ 1 mol%, 0.25M, Doyle et al, *J. Am. Chem. Soc.* 2018, 140, 3035; $(^t\text{Bu}bpy)\text{NiArBr}$ 10 mol% loading, 0.1 M. It is also known that with $(^t\text{Bu}bpy)\text{NiX}_2$ system generates isolable dinuclear Ni(I) species that are not catalytically active: $\text{Ni}_2(^t\text{Bu}bpy)_2(\mu\text{-Cl})_2$ (Doyle: *J. Am. Chem. Soc.* 2020, 142, 5800) or $[\text{Ni}_2(^t\text{Bu}bpy)_2(\text{quinuclidine})_2(\mu\text{-Cl})_2]^+$ (Nocera et al, *J. Am. Chem. Soc.* 2019, 141, 89). In addition, as these dinuclear are not catalytically active, dissociation of the dimer is a prerequisite for the oxidative addition step.

The mononuclear $(^R\text{N}3)\text{Ni}^I$ species we observed in the EPR experiment undergoes oxidative addition in presence of Ar-Br (as shown in Figure 4), suggesting that it is a mononuclear $(^R\text{N}3)\text{Ni}^I$ species and not an inactive dinuclear species. Moreover, the Ni^I species 2 decays rapidly to EPR inactive species in the absence of substrate, yet no dinuclear Ni^I species are formed, suggesting that the tridentate $^R\text{N}3$ ligand might prevent the formation of such catalytically inactive species due to the steric hindrance provided by the axial amine arm of the $^R\text{N}3$ ligand (see Supplementary Figs. S58 and S59 for the space filling models of **1a** and **1b**). This behavior might explain why higher concentrations of substrates and catalysts can be employed herein, which is in contrast to the conditions utilized in other reported C-O cross-coupling reactions.

We thank the reviewer for the insightful suggestion, and we have included this discussion in the manuscript pages 13 and 14.

•The relevance of Ni(I)/(III) comproportionation to Ni(II) needs further discussion. Does **1b**+ CoCp2* (s20) decompose into 4? If so, how does this relate to sustained Ni(I)/(III) catalysis. The low/modest yields in reductive elimination studies in figure 6 mirror previous RE studies from Ni(III) and suggest that comproportionation may limit sustained Ni(I)/(III) catalysis (Nocera, Ref 14). However, figure 4b might suggest otherwise. Overall, I think the present work is highly relevant to previous reports of sustained Ni(I)/(III) catalysis but is not suitably related to these studies. The impact of the work would be more clear if these details were clarified through experiment and explanation.

Our response: We thank the reviewer for this comment. We briefly mentioned the possibility of $\text{Ni}^I/\text{Ni}^{III}$ comproportionation reaction into Ni^{II} in the original version of the Supplementary Information, however we have now also included the discussion in the main text and have performed additional experiments to provide further evidence for such a hypothesis.

As the reviewer suggested, we have performed comproportionation experiments and have included these results and their discussion in the main text page 22. A detailed discussion and experimental details are included in the Supplementary Information page S37. section 10: Comproportionation studies.

To assess the Ni^I and Ni^{III} deactivation pathway, we attempted to experimentally observe the comproportionation between Ni^{III} with in-situ generated Ni^I through EPR (observation of signal decay of **5**) and NMR (observation of $Ni^{II}ArX$ or $Ni^{II}X_2$ species):

EPR spectroscopy: The EPR signal of complex **5** started to decay and completely disappeared within 30 s upon addition of in-situ generated **2** (i.e., **1b** + CoCp*₂), indicating conversion into EPR-inactive species (Figure S25). We note that without the addition of **1b** + CoCp*₂, only marginal signal decay of complex **5** was observed even when the sample was warmed up to room temperature for 5 min (Figure S34).

Figure S1. EPR signal decay of **5** upon addition of in situ generated **2** (i.e., **1b** + CoCp*₂). The initial spectrum of **5** was collected before mixing the sample (black solid line). Each spectrum was recorded at 77 K, after quick thawing to -95 °C for 5 s and subsequent refreezing. The order of the spectra obtained: black-red-blue-green-purple-yellow-dark brown.

NMR spectroscopy:

While the obtained 1H NMR spectrum of the resulting reaction mixture (middle, green spectrum, Figure S26) did not match that of the independently synthesized complex **4** (top, blue spectrum), we observed the formation of organic compounds (Figure S27, bottom: 4,4'-diacetylphenyl, 4-bromoacetophenone, 4-methoxyacetophenone) and complex **1b** (iPr_3N_3)NiCl₂ (Figure S28; the 1H paramagnetic NMR spectrum of independently synthesized **1b** is shown in Figure S10), which could re-enter the catalytic cycle. While we cannot observe complex **4** directly, two peaks (marked with * in Figs. S27 and S29) were observed that are tentatively assigned to a Ni-p-acetylphenyl intermediate that generates acetophenone upon solution workup for GC-MS. We assume that complex **4** is unstable under the reaction conditions, since an

immediate color change from red to yellow was observed for complex **4** in presence of MeCN or MeOH (polar or protic solvent), indicating decomposition or formation of other species. The ^1H NMR of complex **4** obtained upon addition of CD_3OD supports this decomposition or formation of a new species (Figure S29).

Figure S2. ^1H NMR spectra of the complex **4** (top, blue), *in situ* generated **2** (i.e., **1b** + CoCp^*_2 , middle, green), and the reaction mixture after 5 h at RT (bottom, red).

Figure S3. Identification of organic products. Enlarged aromatic region of ^1H NMR spectrum of the reaction mixture (top, purple) and the assignment of organic products: ● = 4,4'-diacetylbiphenyl, ■ = 4-bromoacetophenone, ▲ = 4-methoxyacetophenone, * = peaks likely belonging to complex 4 (or a derivative of), which generates acetophenone upon workup for GC-MS analysis). All ^1H NMR spectra were recorded in $\text{CD}_2\text{Cl}_2 + \text{CD}_3\text{OD}$ under N_2 .

Figure S4. ^1H NMR paramagnetic spectra of complex **4** (top, blue), *in situ* generated **2** (i.e., **1b** + CoCp^*_2 , middle, green), and the reaction mixture left at room temperature for 5 h (bottom-red).

Figure S5. ^1H NMR spectra of the complex **4** (top, blue) in CD_2Cl_2 , the complex **4** in $\text{CD}_2\text{Cl}_2 + \text{CD}_3\text{OD}$ (middle, green), and the reaction mixture left at room temperature for 5 h (bottom, red) with assignment of aromatic peaks: ● = 4,4'-diacetylbiphenyl, * = peaks likely belonging to complex **4** (or a derivative of), which generates acetophenone upon workup for GC-MS analysis.

Overall, while we were not able to observe complex **4**, our EPR and NMR results support the deleterious $\text{Ni}^{\text{I}}/\text{Ni}^{\text{III}}$ comproportionation that leads to off-cycle Ni^{II} dihalide species, which later can re-enter the catalytic cycle upon photoexcitation. This $\text{Ni}^{\text{I}}/\text{Ni}^{\text{III}}$ comproportionation reaction is consistent with previously reported computational (Nocera et al, *J. Am. Chem. Soc.* 2019, 141, 89) or experimental studies (Zargarian et al, *Organometallics* 2018, 37, 1446).

•The NMRs in figure S1 don't appear to be appropriately referenced

Our response: We apologize for the error. We have now properly referenced the stacked NMR spectra along the CD_2Cl_2 peak and have updated figure S1 accordingly.

Reviewer #3 (Remarks to the Author):

In this paper, the authors have developed a new class of tridentate pyridinophane ligands (RN3), with two specific examples of R= H and Me, which when formed into Ni containing complexes act of photocatalysts for C-O coupling reactions. Importantly, the present ligands permit the investigation of the key steps of the catalytic cycle. The Ni complexes are characterized by X-ray diffraction, cyclic voltammetry (CV), ^{13}C and ^1H NMR, and UV-vis absorption. The CV nicely illustrates the possibility of the ligand stabilizing both Ni^{I} and Ni^{III} species. The authors then study the photocatalytic C-O coupling mediated by the (RN3) NiCl_2 catalysts using the reaction of methanol and 4-bromoacetophenone as an example. The authors carry out useful control experiments, and explore conditions for reaction optimization. While these controls and optimizations are important, the more interesting part of the work lies in the reactivity studies, with the present ligands, related complexes, slightly different reaction conditions, or alternate photocatalysts, and their probing with EPR. For example, from their EPR spectra in fig. 4, they suggest that they have directly observed “the oxidation addition of an aryl halide to Ni^{I} species to generate a Ni^{III} species” (see comment 7 below). Overall the experimental work appears to be outstanding and the insight obtained would be of great interest to the community. However, my primary expertise is in computational chemistry, and there are some issues outlined below in the present work. If these can be satisfactorily addressed, and based on the the experimental work, I could tentatively recommend publication.

1. The authors state that their computations have used the M06-tzvp functional. There is no such density functional and I assume the authors mean M06-2X (or M06, or M06-HF or M06-L, or ...). The name of the functional used should be corrected and the authors should include a reference in Section 14 of the SI to the original paper where the functional was first introduced. My rest of my comments are based on the assumption that the M06-2X functional was used in the present work.

Our response: We truly apologize for the confusion, which arose from the including the wrong information about the DFT calculations. We have included the correct details of the computational methods in the revised Supplementary Information.

2. The authors state that they have used the modified 6-31G* (m6-31G*) basis set for nickel. This basis set is not available from the basis set library in Gaussian; therefore, from where did the authors obtain it. They should also reference the original work for this basis set, i.e., The Journal of Chemical Physics 118(17):7775-7782.

Our response: *We truly apologize for the confusion, which arose from the including the wrong information about the DFT calculations. We have included the correct details of the computational methods in the revised Supplementary Information.*

3. The authors justify their choice of functional and basis set by stating “this combination of hybrid functional and basis sets has been previously shown to work well for reproducing experimental parameters for Ni complexes.^{15,16}” Both these references use the B3LYP functional not the one used in the present work, and hence, they cannot be used as justification for the choice used here. Please note that I am not advocating for the use of B3LYP, there are probably other better functionals for the properties (i.e, excited states) considered here, see comment 5.

Our response: *We truly apologize for the confusion, which arose from the including the wrong information about the DFT calculations. We have included the correct details of the computational methods in the revised Supplementary Information.*

4. The authors state that “the ground state of **1a** has triplet spin multiplicity” but no mention is made in the Computational section about defining the spin of the complexes. I assume that this was done, and they were defined as triplets, and thus the question arises as to whether the DFT computations were carried out using unrestricted (default in Gaussian) or restricted formalism. This choice needs to be clearly defined. Moreover, there are significant challenges is using TD-DFT to compute triplet excited states from an unrestricted triplet ground state; in particular, the states computed can be plagued with spin-contamination. Have the authors examined the spin purity of the resultant excited states, especially the two excited states of primary interest?

Our response: *We thank the reviewer for this comment, and apologize for the lack of clarity. The triplet ground state for complexes **1a** and **1b** was confirmed experimentally, as it has been known that magnetic measurements can successfully determine the electronic configurations and multiplicities of the ground state transition metal complexes. (Russell S. Drago, *Physical Methods for Chemists*, 2nd edition, Chapter 11, and Crosby et al, *J. Edu. Chem.* 1983 60 10 791). For example, complex **1b** exhibits a paramagnetic NMR spectrum, and a magnetic moment μ_{eff} of $3.03 \mu_B$ was determined using the Evans method, thus consistent with a $S = 1$ ground state and as expected for a high spin Ni^{II} center. The obtained magnetic moment value is similar to those of well-known triplet ground state $\text{Ni}(\text{bpy})\text{X}_2$ ($\text{X} = \text{halide}$) complexes (Broomhead and Dwyer et al, *Australian Journal of Chemistry*, 1961, 14(2) 250). We have added a statement in the main text pages 6–7 and Supplementary Information page S6. We have also investigated in more detail the spin contamination of the excited states of interest and they were confirmed to be triplet excited states. A statement was added on Supplementary Information page S72.*

5. The authors have assigned the two primary excitations to CT excitations (at 380 nm) and a metal-centred transition (at 640 nm). There are important considerations about the choice of functional for CT

transitions; for many hybrid functionals, their use leads to artificially low excitation energies for CT states, and thus energetic ordering is incorrect, see e.g. J Chem Phys. 2008 Jan 28;128(4):044118.

I wonder about the extremely low-energy states (< 1 eV) seen in the long list of computed excited states, see Table S19 and S23 where 100 excited triplet (??) states are reported; note, it is extremely unlikely for TD-DFT to capture this many states correctly although it may provide a reasonable representation for the important low-lying bright states. These CT states can be accounted for by using a long-range corrected functional, e.g. ω B97xD, CAM-B3LYP. It would be useful for the authors to confirm their state assignments using one of these functionals.

Our response: We thank the reviewer for this comment, and we apologize for the lack of clarity. We have performed additional TD-DFT calculations using the M06 functional, the long-range corrected functionals CAM-B3LYP and ω B97XD, and the def2tzvp basis set were also employed to probe their effect on the TD-DFT calculated UV-vis spectra and the nature of the electronic transitions (Figure S72). For all these simulated UV-vis spectra, while the predicted transition energies vary slightly depending on the functional/basis set combination, inspection of the NTOs corresponding to the transitions at 300-400 nm and 450-600 nm reveal that these are mainly charge transfer and d-d transitions, respectively, for all functional/basis set combinations.

Figure S6. Calculated UV-vis spectra for **1b** based on TD-DFT calculations employing various functional/basis set combinations. While the predicted transition energies vary slightly depending on the functional/basis set combination, inspection of the NTOs corresponding to the transitions at 300-400 nm and 450-600 nm reveal that these are mainly charge transfer and d-d transitions, respectively, for all functional/basis set combinations.

6. On page 8, the authors write that “the MC(d-d) state is the lowest excited state” but their TD-DFT results (Tables S19 and S23) show 4 lower-lying excited states, see also comment 5. The authors also refer to “the higher energy CT state and MC(d-d) state are close in energy, thus resulting in fast relaxation to the lowest lying MC(d-d) state.” However, these states differ in energy by greater than 1 eV; this is not close so does not seem to support the statement.

Our response: We thank the reviewer for this comment. We agree with the reviewer that that there are indeed lower energy transitions are also MC(d-d) transitions, yet they do not involve Ni–Cl σ^* orbital population. In addition, such transitions with small oscillator strengths are less likely related to the photochemistry that we are interested in and therefore were not considered. However, we agree that the excited states we investigated in more detail are not the lowest excited state, therefore we have reworded the corresponding sentence on page 9 of the main text to remove any confusion.

In addition, we are referring to the MLCT/LLCT state (Excited state 1) that lies closer in energy to the lower-lying MC(d-d) excited state (Excited state 2) than the ground state (as shown in the image below). Even though the gap between the MLCT/LLCT state and the MC(d-d) state is greater than 1 eV, the gap between the MLCT/LLCT excited state and the ground state is much larger, thus there is more probability of relaxation into MC (d-d) state and then deactivation (decay2) compared to the direct deactivation into the ground state (decay 1). Since we believe that our original sentence led to the confusion, we have modified the text on page 9 and removed the word “close”.

7. The authors propose the presence of a 6-coordinate NiIII species (3) on the basis of EPR measurements. Have the authors considered computing the EPR g-tensor to try and confirm the presence of this species?

Our response: We thank the reviewer for this comment. Unfortunately, since we don't have any structural information on species 3, it would have taken too long to screen many potential geometries and conformations, optimize those geometries, and then calculate relevant EPR parameters. However, we have calculated the EPR parameters for the more stable species 5 (for which we have a reasonable guess structure, similar to the Ni^{II} precursor 1b), and the calculated parameters are closely matching the experimental values (Table S11). Thus, in future studies we will employ DFT-calculated EPR parameters to provide additional support for the geometry of transient paramagnetic species.

Minor points

8. For the simulated UV-vis spectra, the line shape and width for each of the contributing peaks should be provided.

Our response: We thank the reviewer for this comment. Gaussian peak shapes were used with a full width at half maximum (FWHM) of 0.33 eV. We provided this information in the Figure 3(a) caption and also in the Supplementary Information page S72: Section 17. DFT calculations.

REVIEWERS' COMMENTS

Reviewer #1 (Remarks to the Author):

The authors have genuinely responded to the comments with a significant body of new work. Their response speaks the high level of the author's scholarship. This effort is appreciated. With regard to the points raised in the initial review of this manuscript:

- (1) The inclusion of the UV-vis details in S24 is welcome, and the argument of interfering absorption of reagents and low absorption of complex justifies the lack of reactivity with blue LED excitation.
- (2) The "on-off" cycle experiments do support a dark cycle, and it is appreciated the care in which the authors have performed additional experiments. It is unusual though that the quantum yield is < 1 , as a dark cycle would greatly increase the quantum yield ... i.e., many turnovers (dark reaction) per photon. The authors may wish to comment on why a contributing dark reaction does not lead to a $qy > 1$, as I believe many readers of the manuscript will be interested in a possible rationalization. There are two obvious reasons: (1) side decomposition reactions that mitigate the $qy > 1$ or that the qy for photoexcitation is $<< 1$ and the dark reaction qy cannot overcome mostly thermal nonradiative processes. If they have a sense of the possibility, a few lines might be warranted. But I also would understand if the authors were reticent to be too speculative. Thus a few lines of possibilities might be in order? I leave it to the author's discretion on how they would like to present their work on this subject in the manuscript.
- (3) Ok.
- (4) The added discussion on this point is appreciated.
- (5) I concur with the editorial comment and regret in forgetting to include this in my original review (vs a subsequent comment to the Editor after the initial review).

Reviewer #2 (Remarks to the Author):

The authors have satisfactorily addressed the concerns. Publish as is.

Reviewer #3 (Remarks to the Author):

As my primary expertise is in computational chemistry, I will comment on the revisions to the reported computational work. In general, the authors have clarified and addressed most of the questions/comments raised but a few minor points remain. The authors have utilized several functionals (two hybrids: B3LYP and M06, as well as two range-separated hybrids CAM-B3YP, and ω B97xD, see point 3 below) and the tzvp and def2-tzvp basis sets for the excited state computations [Note that one would expect results with these two basis sets to be very similar, as def2-tzvp is simply a revision of tzvp]. While there are changes in energies and oscillator strengths between functionals, the authors state that the nature of the low-lying excitations do not change significantly; note that this is not demonstrated explicitly as the only figure comparing functionals is S72 presenting the UV-Vis spectra. However, it is important that the qualitative interpretation of the nature of the excitations does not change. Minor points:

1. In the Figures in the SI (S60, S61, S62, S63, S64, S65, S66, S67, S68, S69, S70, S71) as well as Tables (S17, S18, S19, S21, S22, S23) the authors should explicitly provide the functional/basis set used for the presented results. Comparing Figure S60 to Figure S72, it appears that the "simulated" results in S60 are for M06/tzvp, but in all other figures/tables, the choice of functional/basis set cannot be determined.
2. Comparing Tables S18 and S19, as well as Table S22 and S23, there appears to be some truncation/rounding in the latter tables as the numbers in both tables do not precisely agree (i.e., the remaining 8 zeros in Tables S19 and S3 are meaningless).
3. The functional is ω B97xD not wB97xD, i.e, it is an omega not the letter "w."

4. The authors mention that they have now examined the spin purity of the resultant excited states, including the two excited states of primary interest, and that they were "confirmed to be triplet excited states." It would be useful to report the resulting S^2 values.

REVIEWER COMMENTS

REVIEWER #1

The authors have genuinely responded to the comments with a significant body of new work. Their response speaks the high level of the author's scholarship. This effort is appreciated. With regard to the points raised in the initial review of this manuscript:

(1) The inclusion of the UV-vis details in S24 is welcome, and the argument of interfering absorption of reagents and low absorption of complex justifies the lack of reactivity with blue LED excitation.

(3) Ok.

(4) The added discussion on this point is appreciated.

(5) I concur with the editorial comment and regret in forgetting to include this in my original review (vs a subsequent comment to the Editor after the initial review).

Our response to (1), (3), (4), and (5): We appreciate the reviewer for all current and previous comments and suggestions.

(2) The “on-off” cycle experiments do support a dark cycle, and it is appreciated the care in which the authors have performed additional experiments. It is unusual though that the quantum yield is < 1 , as a dark cycle would greatly increase the quantum yield ... i.e., many turnovers (dark reaction) per photon. The authors may wish to comment on why a contributing dark reaction does not lead to a $qy > 1$, as I believe many readers of the manuscript will be interested in a possible rationalization. There are two obvious reasons: (1) side decomposition reactions that mitigate the $qy > 1$ or that the qy for photoexcitation is $\ll 1$ and the dark reaction qy cannot overcome mostly thermal nonradiative processes. If they have a sense of the possibility, a few lines might be warranted. But I also would understand if the authors were reticent to be too speculative. Thus a few lines of possibilities might be in order? I leave it to the author’s discretion on how they would like to present their work on this subject in the manuscript.

Our response: We thank the reviewer for this comment. We agree with the reviewer’s comment regarding the low QY. In our manuscript, we experimentally confirmed that Ni^I/Ni^{III} comproportionation reaction can generate off-cycle Ni^{II} species. In addition, several other deleterious pathways could be active, such as Cl radical capture by the Ni^I species to regenerate Ni^{II} . We also consider that our QY of 0.26 determined by actinometry could be significantly overestimated. In such a case, if the QY $\ll 1$, then the dark catalytic cycle may not overcome the thermal nonradiative processes, as the reviewer has suggested.

Therefore, we have included an additional explanation in the main text p21, and also in the SI, pS33: “In addition, the quantum yield ($\Phi_{Ni} < 0.26$) obtained by actinometry could be significantly overestimated. Given that there are several possible side reactions that would generate off-cycle Ni^{II} species, if $\Phi_{Ni} \ll 1$, then the dark Ni^I/Ni^{III} dark catalytic cycle may not be able to compensate for the deleterious side reactions.”

REVIEWER #2

The authors have satisfactorily addressed the concerns. Publish as is.

Our response: We appreciate the reviewer’s recommendation and want to thank the reviewer for all previous comments and suggestions.

REVIEWER #3

As my primary expertise is in computational chemistry, I will comment on the revisions to the reported computational work. In general, the authors have clarified and addressed most of the questions/comments raised but a few minor points remain. The authors have utilized several functionals (two hybrids: B3LYP and M06, as well as two range-separated hybrids CAM-B3YP, and ω B97xD, see point 3 below) and the tzvp and def2-tzvp basis sets for the excited state computations [Note that one would expect results with

these two basis sets to be very similar, as def2-tzvp is simply a revision of tzvp]. While there are changes in energies and oscillator strengths between functionals, the authors state that the nature of the low-lying excitations do not change significantly; note that this is not demonstrated explicitly as the only figure comparing functionals is S72 presenting the UV-Vis spectra. However, it is important that the qualitative interpretation of the nature of the excitations does not change.

Our response: *We thank the reviewer for this comment. We have included in Supp. Figs 74–78 the nature of corresponding transitions determined by TD-DFT calculations employing various functional/basis set combinations. Comparison of those figures with Supp Fig 65 confirms that the nature of the transitions (the higher energy MLCT/LLCT transition and the lower energy MC(d-d) transition) does not change depending on the choice of functional/basis set.*

Minor points:

1. In the Figures in the SI (S60, S61, S62, S63, S64, S65, S66, S67, S68, S69, S70, S71) as well as Tables (S17, S18, S19, S21, S22, S23) the authors should explicitly provide the functional/basis set used for the presented results. Comparing Figure S60 to Figure S72, it appears that the “simulated” results in S60 are for M06/tzvp, but in all other figures/tables, the choice of functional/basis set cannot be determined.

Our response: *We thank the reviewer for this comment. As suggested, we have provided the functional/basis set information in each figure and table captions in Supplementary Figs. 61–73, and Supplementary Table 16–22 (Note that Figure and Table numbers have changed, originally there were Figures S60–S72 and Tables S17–S23).*

2. Comparing Tables S18 and S19, as well as Table S22 and S23, there appears to be some truncation/rounding in the latter tables as the numbers in both tables do not precisely agree (i.e., the remaining 8 zeros in Tables S19 and S23 are meaningless).

Our response: *We apologize for the error, we have corrected the data in Supp Tables S18 and S22, which have been renumbered Supplementary Table 17 and 21, respectively. The disagreement of the numbers in those tables was due to the precision number and increment setup in the Chemission software while producing the Tables. We also removed the meaningless zeros in Tables S19 and S23 (renumbered Supplementary Table 22 and 18 currently).*

3. The functional is ω B97xD not wB97xD, i.e, it is an omega not the letter “w.”

Our response: *We apologize for the typo, we have corrected Figure S72 (Supplementary Figure 73 currently) and the corresponding text.*

4. The authors mention that they have now examined the spin purity of the resultant excited states, including the two excited states of primary interest, and that they were “confirmed to be triplet excited states.” It would be useful to report the resulting $\langle S^2 \rangle$ values.

Our response: *We thank the reviewer for this comment. We have provided the $\langle S^2 \rangle$ values in Supplementary Tables 17 and 21 as suggested.*